# *AugServe*: Adaptive Request Scheduling for Augmented Large Language Model Inference Serving

**Ying Wang** [1]   **Zhen Jin** [1]   **Zhenqian Chen** [1]   **Jiexiong Xu** [1]   **Wenhai Lin** [2]   **Yiquan Chen** [2]   **Wenzhi Chen** [1]

## Abstract

Augmented large language models (LLMs) that invoke external calls are increasingly prevalent in inference serving. However, such augmentations pose significant challenges to inference efficiency under strict Service-Level Objectives (SLOs). Existing inference systems are agnostic to the dynamic execution behaviors induced by external calls and rely on fixed batch-level token budget, which leads to severe Head-of-Line (HoL) blocking and substantially reduced effective throughput. We present *AugServe*, an efficient augmented LLM inference serving framework that mitigates request queuing latency and improves effective throughput under external-call-augmented workloads. *AugServe* integrates state-aware request scheduling with dynamic batch-level token budgets to adapt to heterogeneous requests and their dynamically changing execution states. Experimental results show that *AugServe* achieves $6.5\times$ and $4.7\times$ higher effective throughput than vLLM and INFERCEPT, respectively.

## 1. Introduction

Augmented Large Language Models (LLMs) have rapidly emerged as a promising paradigm (Abhyankar et al., 2024; Hao et al., 2023) for modern LLM inference serving. Compared with traditional text-only LLMs, which rely on fixed pretrained parameters and lack real-time knowledge (Schick et al., 2023; Gade et al., 2025), augmented LLMs extend their capabilities by invoking external tools (e.g., web APIs, databases, or specialized models) during inference (Chen et al., 2024b; Go & Park, 2025; Qin et al., 2024a; Lu et al., 2024). This approach enables augmented LLMs to perform

[1]College of Computer Science and Technology, Zhejiang University, Hangzhou, China [2]Alibaba Group, Hangzhou, China. Correspondence to: Jiexiong Xu <jasonxu@zju.edu.cn>, Wenzhi Chen <chenwz@zju.edu.cn>.

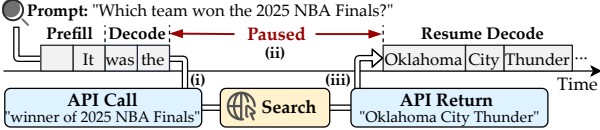

*Figure 1.* Augmented LLM inference process.

more complex tasks such as arithmetic computation (Chen et al., 2024a; Yao & Yadav, 2025), real-time information retrieval (Su et al., 2024; Gade et al., 2025), and web interactions (Qi et al., 2025; Zhang et al., 2025).

Augmented LLM inference service systems are becoming the key infrastructure for AI-centric cloud computing, with inference efficiency directly impacting user experience. Figure 1 illustrates the workflow of the augmented LLM inference service (Abhyankar et al., 2024; Gim et al., 2024): **(i)** During inference, the augmented LLM identifies the need for real-time information and triggers the corresponding tool calls. **(ii)** The inference process is paused while awaiting the response from the external augmentation module. **(iii)** Upon the response being returned, the serving system appends it to the sequence generated and resumes normal generation.

Ideally, inference systems must simultaneously deliver high throughput and low latency. In this context, **Service-Level Objectives (SLOs)** serve as strict latency boundaries (e.g., requiring Time-to-First-Token (TTFT) below a fixed threshold) (Gao et al., 2025; Zhong et al., 2024; Wu et al., 2023; Patel et al., 2025). Accordingly, the system's efficiency is best characterized by **effective throughput** (or **goodput**), defined as the volume of requests processed per unit time that successfully satisfy these SLO requirements (Wang et al., 2025; Karthik et al., 2024; Zhang et al., 2023).

State-of-the-art inference systems focus on improving inference performance. vLLM (Kwon et al., 2023) has emerged as the de facto standard for efficient LLM serving. However, in augmented LLM inference, vLLM treats external calls as request termination and discards the request's context (i.e., Key-Value (KV) cache) (Abhyankar et al., 2024). When the call returns, the system must recompute the KV cache, incurring substantial computation overhead and processing latency. To address this issue, INFERCEPT (Abhyankar

et al., 2024) dynamically manages context based on call duration and context length, selecting among three KV-cache handling policies: discarding it, preserving it in GPU memory, or swapping it to the host memory. This design reduces resource waste and significantly improves the efficiency of augmented LLM inference.

However, these systems still face two challenges in improving the goodput of augmented LLM inference:

**C1: Inadequate scheduling leads to Head-of-Line (HoL) blocking and SLO violations.** In augmented LLM inference, existing systems (e.g., vLLM and INFERCEPT) typically adopt First-Come-First-Served (FCFS) scheduling, batching requests without accounting for external calls. When long requests trigger external calls and pause execution, their context (KV cache) may be preserved in GPU memory, swapped out to host memory, or discarded, all of which can block subsequent short requests. This results in severe HoL blocking, causing queuing delays that exceed SLOs and sharply degrade goodput. Some work attempts to mitigate these delays using approximate Shortest-Job-First (SJF) scheduling based on request length (Jin et al., 2023; Fu et al., 2024; Wu et al., 2023). However, they still ignore the execution heterogeneity introduced by external calls and remain suboptimal for augmented LLM inference.

**C2: Fixed batch-level token budget restricts throughput under external calls.** In augmented LLM inference, external calls introduce paused requests whose contexts may occupy GPU memory, complicating batch capacity selection. A static batch-level token budget cannot adapt to this dynamic memory availability. A small budget limits per-iteration concurrency and reduces throughput, while an overly large budget induces resource contention and frequent eviction of paused contexts, incurring recomputation overhead. Moreover, existing approaches (Zheng et al., 2025) adjust the budget based only on free GPU memory, ignoring reclaimable memory under different context-handling policies, leading to suboptimal budget decisions.

In this paper, we propose *AugServe*, an augmented LLM inference serving framework that jointly rethinks request ordering and batch capacity. Our key insight is that augmented LLM inference introduces significant execution heterogeneity, where requests with different external calls and context-handling policies exhibit distinct resource demands across multiple stages. Guided by this insight, *AugServe* adopts a unified, state-aware design that optimizes request scheduling and batch-level capacity adaptation to maximize goodput under dynamic augmented LLM workloads.

For **state-aware scheduling (C1)**, *AugServe* models request scheduling across the full inference lifecycle, explicitly capturing multi-stage execution and cross-round state evolution induced by external calls. Scheduling priorities are con-

structed in a state-aware manner that adapts to each request's execution state, context-handling outcome, and observed runtime feedback. By prioritizing requests with higher execution efficiency given their current execution stages and resource footprint, *AugServe* alleviates HoL blocking in augmented LLM inference, significantly reducing queuing delays and improving goodput.

For **dynamic batch-level token budget (C2)**, *AugServe* adapts batch capacity based on available GPU memory and reclaimable memory from paused requests under different context-handling policies. Besides, *AugServe* enforces bounded budget adjustments to ensure robustness.

We implemented *AugServe* with vLLM and evaluated it against both vLLM and INFERCEPT across multiple LLMs and GPU platforms. Experimental results show that *AugServe* consistently outperforms both baselines in latency and effective throughput. In particular, *AugServe* achieves a geometric mean of $6.5\times$ and $4.7\times$ higher effective throughput than vLLM and INFERCEPT, respectively, while reducing TTFT 95.6% and 96.0% on average.

In summary, our contributions are as follows:

- We present *AugServe*, an augmented LLM inference serving framework that efficiently improves effective throughput (§5).
- We propose an adaptively state-aware request scheduling strategy that optimizes request ordering based on request characteristics, external-call-induced execution states, and runtime feedback. (§5.2, §5.3)
- We develop a dynamic batch-level token budget mechanism adapting to free and reclaimable memory. (§5.4)
- We conduct extensive evaluations to validate the effectiveness of *AugServe* (§6).

## 2. Background

We review augmented LLMs and existing inference systems.

### 2.1. Augmented Large Language Models

Augmented LLMs integrate external tools (e.g., remote APIs, databases, external models) during inference (Chen et al., 2024b; Schick et al., 2023; Lu et al., 2024; Go & Park, 2025), demonstrating superior performance in complex tasks such as arithmetic computation (Hao et al., 2023; Chen et al., 2024a; Yao & Yadav, 2025) and real-time information retrieval (Su et al., 2024; Gade et al., 2025). Furthermore, with the rise of tool-using agents (Patil et al., 2025; Wölflein et al., 2025) and the standardized tool interactions via the Model Context Protocol (MCP) (Anthropic, 2024), tool invocation has become a ubiquitous component of inference pipelines (Fei et al., 2025; Mialon et al., 2023). Consequently, augmented LLM inference systems are emerging as

the core infrastructure for next-generation cloud platforms.

## 2.2. Existing LLM Inference Systems

LLM inference has become a dominant workload in modern data centers, motivating the design of advanced systems to improve overall efficiency. To handle varying request sequence lengths, Orca (Yu et al., 2022) introduces iteration-level scheduling, which has become the de facto standard in state-of-the-art inference engines. Meanwhile, to improve GPU memory utilization, vLLM (Kwon et al., 2023) proposes PagedAttention to eliminate memory fragmentation. Additionally, some research explores offloading KV cache to CPU or SSD (Jeong & Ahn, 2025; Hu et al., 2025b; Sheng et al., 2023) to alleviate GPU memory bottlenecks.

However, in augmented LLM inference, most prior works simply discard the context (KV cache) during external calls. INFERCEPT (Abhyankar et al., 2024) improves over these approaches by dynamically selecting among `Preserve`, `Discard`, or `Swap` context-handling policies based on the external call duration and context length:

- `Preserve`: The KV cache remains in GPU memory, and decoding resumes once the response returns.
- `Discard`: The KV cache is discarded, and recomputation is required after the response returns.
- `Swap`: The KV cache is swapped to CPU memory and restored to GPU memory once the response returns.

This adaptive design avoids inefficient reliance on a single policy, reducing memory waste and inference latency.

## 3. Motivation

Maximizing goodput under SLOs is challenging in augmented LLM inference, where requests are heterogeneous and involve multi-round external calls. This section analyzes the key limitations of existing approaches.

### 3.1. Limitations of Existing Scheduling Strategies

**Challenge 1: Existing inference scheduling strategies struggle in dynamic augmented LLM serving.** Current inference systems typically adopt FCFS scheduling and batch requests by arrival order, without considering external calls. In augmented LLM inference, pausing requests for external calls (whether context is preserved, swapped, or discarded) blocks subsequent short queries. This induces severe HoL blocking, increasing latency and degrading goodput. To alleviate HoL blocking, prior work explores approximate SJF scheduling based on request length (Jin et al., 2023; Fu et al., 2024; Wu et al., 2023). While these approaches outperform FCFS (Figure 2), they do not account for external calls and still fail to meet SLOs in augmented LLM inference, with TTFT often exceeding the SLO (e.g., 1s), high tail latency,

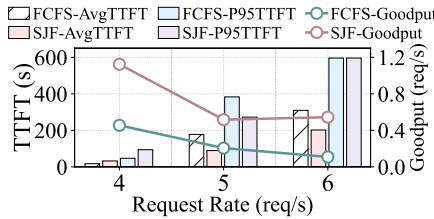

*Figure 2.* SJF outperforms FCFS in goodput (effective throughput) and TTFT, but remains suboptimal under high load.

and degraded goodput at high load. Recent systems (Shahout et al., 2025b) propose memory-based SJF heuristics that approximate job sizes using memory cost. However, these approaches make one-shot scheduling decisions at each execution round, treating requests as newly arriving jobs after external calls. Consequently, they do not model dynamic and cross-round execution stages in augmented LLM workloads (Figure 5), ignoring resumption costs from cumulative context and external call return lengths.

Furthermore, we observe that both the cumulative context lengths at the time of external calls and the external call return lengths are highly variable (Figure 3). Such variability results in dynamic resumption costs after external calls under different context-handling policies, which have distinct memory footprints and recomputation overheads. Moreover, the varying external call return lengths significantly affect both TTFT and goodput (Figure 4). Together, these factors make existing scheduling policies suboptimal for augmented LLM inference.

**Opportunity 1:** Augmented LLM inference calls for scheduling mechanisms that explicitly account for heterogeneous execution states induced by external calls and dynamic resumption overhead.

### 3.2. Fixed Batch-Level Token Budget

**Challenge 2: Fixed batch-level token budget restricts throughput under external calls.** The batch-level token budget determines the maximum number of tokens processed in a single forward iteration. In augmented LLM inference, external calls introduce paused requests whose contexts may occupy GPU memory, fundamentally complicating budget selection. While a small budget leads to low per-iteration concurrency and reduced goodput, an overly large budget can trigger resource contention and unnecessary eviction of paused request contexts, incurring redundant recomputation overhead. Recent systems (Zheng et al., 2025) adjust the budget based on free GPU memory, but do not model reclaimable memory from paused requests under `Swap` or `Discard` context-handling policies, failing to capture the true memory availability under external-call-augmented workloads and constraining goodput.

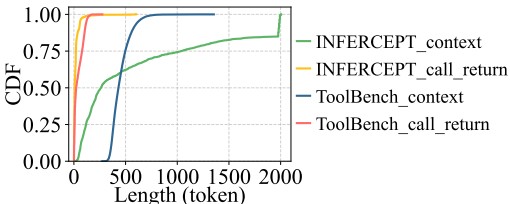

*Figure 3.* Context/External_call_return length distribution in INFERCEPT and ToolBench datasets.

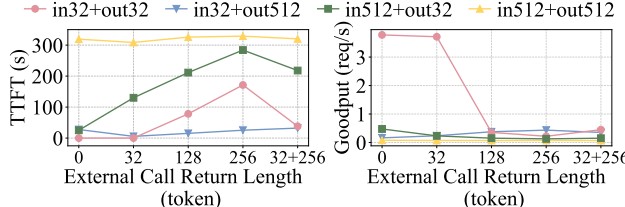

*Figure 4.* TTFT and goodput with varying external call return lengths, including fixed lengths and a mix of 32- and 256- tokens.

*Table 1.* Goodput (req/s) with different batch-level token budgets, optimal token budget varies across hardware and workloads.

| Max batch tokens | 100 | 500 | 1000 | 1500 | 2000 |
|---|---|---|---|---|---|
| 2req/s, GPT-J-6B, RTX4090 | 0.29 | **0.41** | 0.25 | 0.11 | 0.10 |
| 4req/s, OPT-13B, H800 | 0.01 | 0.16 | 0.18 | **0.22** | 0.19 |

We evaluate statically configured batch-level token budgets across different hardware, models, and workloads, and observe that both overly small and excessively large budgets degrade goodput (Table 1). Moreover, the optimal budget varies across hardware and workload conditions.

**Opportunity 2:** Dynamically adapting the batch-level token budget based on both free and reclaimable GPU memory from paused requests is crucial for maximizing goodput in augmented LLM inference.

## 4. Problem Formulation

To systematize request scheduling in augmented LLM inference, this section formalizes the request lifecycle and establishes the associated scheduling objectives.

**Request Lifecycle Modeling.** As shown in Figure 5, in augmented LLM inference, a request spans multiple execution stages (e.g., prefill and decode) interleaved with external call waiting stages. The response length returned by external calls determines the input size and KV cache state for the subsequent execution phase. Accordingly, we model each request $R_i$ as a sequence of $n$ service segments, i.e., $R_i = \{S_{i,1}, \ldots, S_{i,n}\}$, each ending with an external call. Formally, the $k$-th segment $S_{i,k}$ is defined as:

$$S_{i,k} = \begin{cases} \langle P_{i,1}, D_{i,1}, W_{i,1} \rangle, & k = 1, \\ \langle R_{i,k}^{res}(p_i), P_{i,k}^{ret}, D_{i,k}, W_{i,k} \rangle, & 2 \leq k \leq n. \end{cases} \quad (1)$$

The initial segment $S_{i,1}$ comprises standard prompt prefill stage $P_{i,1}$, decoding stage $D_{i,1}$, and a tool-wait stage $W_{i,1}$. Subsequent segments ($k \geq 2$) introduce a context resumption stage $R_{i,k}^{res}$, where $res$ denotes resuming a paused request under a context-handling policy $p_i \in \Pi =$

*Figure 5.* Augmented LLM inference request lifecycle modeling.

{Preserve, Swap, Discard}. This is followed by an incremental prefill stage $P_{i,k}^{ret}$, where $ret$ denotes incorporating tool return tokens into the KV cache. This segmentation treats each segment as a composite unit with distinct, state-dependent resource requirements, providing the necessary granularity for state-aware scheduling.

**Scheduling Objective.** Our goal is to maximize goodput in augmented LLM inference, accounting for both request service rates and queuing delays. Accordingly, we jointly consider how many requests can be served per iteration and how much waiting time can be reduced. To quantify the per-iteration benefit of scheduling a request, we define a *scheduling value* $V_i^e$ capturing both its throughput contribution and waiting-time reduction:

$$V_i^e = 1 + \beta \cdot w_i^e, \quad (2)$$

where the constant term 1 represents the fundamental contribution of serving a request to system throughput progress, $w_i^e$ denotes the waiting time that would be eliminated if request $i$ is scheduled at iteration $e$. The hyperparameter $\beta$ balances throughput-oriented scheduling and delay awareness. A smaller $\beta$ prioritizes throughput by allowing the throughput term to dominate $V_i^e$, whereas a larger $\beta$ shifts the scheduling focus toward reduced waiting time.

Given the candidate request set $U^e$, let $\mathbf{x}^e = \{x_i^e \mid i \in U^e\}$ denote the batch selection decisions at iteration $e$, where $x_i^e \in \{0, 1\}$ indicates whether request $i$ is selected for execution. The scheduling objective is to maximize the aggregate scheduling value of the selected batch, subject to the available GPU memory budget $M^e$:

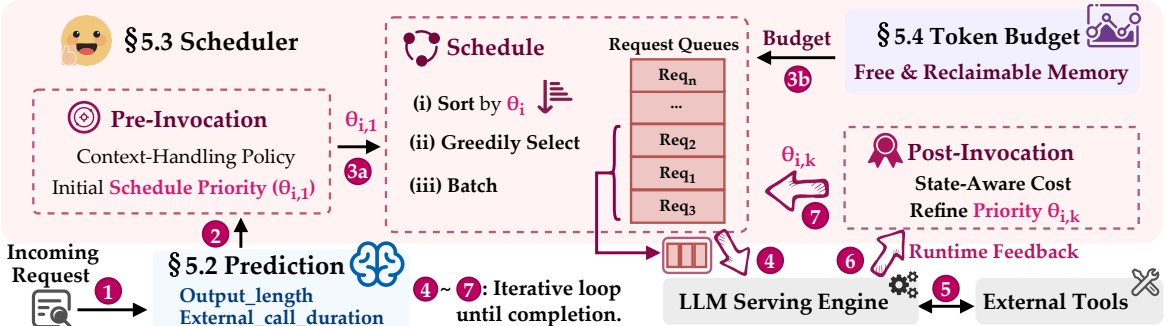

*Figure 6.* System architecture of *AugServe* with state-aware scheduling and dynamic batch-level token budget.

$$\text{Maximize} \quad \sum_{i \in U^e} x_i^e \cdot V_i^e$$
$$\text{s.t.} \quad \sum_{i \in U^e} x_i^e \cdot m_i(t) \leq M^e. \tag{3}$$

## 5. Design

This section details the design of *AugServe*, a unified inference framework for augmented LLM serving.

### 5.1. Design Overview

As illustrated in Figure 6, *AugServe* consists of three modules: (1) **Prediction module** (§5.2) estimates *output_length* and *external_call_duration* to provide priors for scheduling. (2) **Scheduler module** (§5.3) employs a state-aware scheduling policy that incorporates external call characteristics and runtime feedback to optimize request ordering. (3) **Token budget module** (§5.4) adjusts the batch-level token budget based on free GPU memory and preemptable paused-request contexts. Together, these modules form a unified framework that reduces queuing latency and maximizes goodput for augmented LLM inference.

The overall workflow proceeds as follows: An incoming request (①) is first processed by the prediction module to estimate *output_length* and *external_call_duration* as scheduling priors (②). The scheduler utilizes these to determine the context-handling policy and assign an initial priority (③a). Concurrently, the token budget module sets a batch-level token budget based on available GPU memory and reclaimable memory from preemptable paused requests (③b). Then the scheduler ranks requests by priority $\theta_i$ and selects requests to form an execution batch under the token constraint (④). If an external tool is invoked, inference is paused until the call returns (⑤), after which runtime feedback is used to update the request's scheduling priority (⑥⑦). Requests are iteratively scheduled until completion.

### 5.2. Lightweight Request Prediction

To support scheduling, we fine-tune a lightweight BERT-base model (110M parameters) to jointly estimate *output_length* and *external_call_duration*, which provide coarse-grained priors about each request's future execution behavior. *Output_length* prediction is formulated as a classification task by discretizing sequences into fixed 50-token buckets for robustness (Hu et al., 2025a; Jin et al., 2023). Using request inputs as features and realized output lengths as labels, the model achieves 85% bucket accuracy on ToolBench and 65% on the INFERCEPT dataset. *External_call_duration* is predicted via regression using request input metadata as features and actual execution times as labels, with mean squared errors of approximately 5s on ToolBench and 0.4s on INFERCEPT. In addition, the predictor is invoked upon request arrival and tool call return, running off the critical execution path, and adds less than 0.1% overhead to total execution time (§6.2). Importantly, inaccuracies in early predictions are mitigated by continuously refined scheduling decisions (§5.3). We empirically validate this robustness to prediction errors in §6.3.

### 5.3. Adaptive Scheduling with State Awareness

**Value-Density-Based Scheduling Policy.** The scheduling task defined in Equation 3 is a variant of the NP-hard 0-1 knapsack problem, aiming to pack the maximum aggregate value into limited GPU memory. To address this, we propose a greedy value-density strategy that ranks requests by their scheduling efficiency, prioritizing those that yield **higher scheduling value (Equation 2) per unit of resource consumption**. We define $C_{i,k}$ as the resource consumption of the $k$-th service segment of request $i$. The scheduling priority (value density) is then computed as:

$$\theta_{i,k} = \frac{V_i}{C_{i,k}} = \frac{1 + \beta \cdot w_i}{C_{i,k}}. \tag{4}$$

Here, scheduling value $V_i$ captures a request's contribution to throughput progress and waiting-time reduction, computed by tracking the elapsed waiting time $w_i$ since its last

scheduling. This density-based ranking prioritizes requests with higher scheduling efficiency, enabling the scheduler to maximize aggregate throughput while minimizing queuing delays. The remaining key challenge lies in accurately modeling the state-dependent resource consumption $C_{i,k}$ across heterogeneous execution phases, which we address next.

**State-Aware Resource Consumption Modeling.** In augmented LLM serving, resource consumption depends not only on request length but also on external call behavior and context-handling policies. External calls induce variable pause durations and response sizes, while different context-handling policies incur distinct memory and recomputation costs. In addition, execution costs vary across service stages, including prefill, decoding, waiting, and resumption. These factors make length-based or instantaneous memory-based cost models inadequate. We therefore adopt a space-time cost model that quantifies resource consumption as the integral of memory occupancy over the segment's service duration. Formally, the space-time cost of service segment $S_{i,k}$ under context-handling policy $p_i$ is defined as:

$$C_{i,k}(p_i) = \int_0^{\tau_{i,k}} m_i(t, p_i)\, dt, \quad (5)$$

where $m_i(t, p_i)$ denotes the instantaneous memory occupancy of request $i$ under policy $p_i$, and $\tau_{i,k}$ is the segment's residency duration. This formulation measures how long and how much GPU memory a request occupies over its service duration, providing a unified metric to evaluate resource overhead across heterogeneous execution stages. To enable effective scheduling, we further construct $C_{i,k}$ in a state-aware manner, conditioned on the execution state, context-handling policy, and realized external call outcomes. This construction proceeds in two regimes, including Pre-Invocation estimation and Post-Invocation refinement.

*(1) Pre-Invocation Estimation:* Upon request arrival, *AugServe* estimates the space-time cost $C_{i,1}$ to compute the initial scheduling priority $\theta_{i,1}$ before the external call returns. Specifically, a lightweight prediction module (§5.2) provides the predicted *external_call_duration* $\hat{\tau}_i^{call}$ and *output_length* $\hat{l}_i^{out}$ [1]. With these predictions, *AugServe* selects an initial context-handling policy $\hat{p}_{i,1}$ that minimizes the expected memory waste $\text{Waste}_i^{p_i}$ caused by the external call:

$$\hat{p}_{i,1} = \arg\min_{p \in \Pi} \text{Waste}_i^p(\hat{l}_i^{out}, \hat{\tau}_i^{call}). \quad (6)$$

Detailed formulations and symbol definitions are provided in Appendix B. Next, *AugServe* constructs the cost $C_{i,1}$ using Equation 5, combining: (i) the execution cost during the prefill $C_{i,1}^{prefill}$ and decoding stages $C_{i,1}^{decode}$ ($P_{i,1}$ and $D_{i,1}$ in Figure 5), and (ii) the memory residency cost $C_{i,1}^{call}$

---

[1]We use $\hat{\cdot}$ to denote predicted values throughout the paper.

during the external call under policy $\hat{p}_{i,1}$ ($W_{i,1}$ in Figure 5):

$$C_{i,1}(\hat{p}_{i,1}) = C_{i,1}^{prefill} + C_{i,1}^{decode} + C_{i,1}^{call}(\hat{p}_{i,1}) \quad (7)$$

$$C_{i,1}^{call}(\hat{p}_{i,1}) = \begin{cases} C_{i,1}^{preserve} & \hat{p}_{i,1} = \texttt{Preserve} \\ 0 & \hat{p}_{i,1} = \texttt{Discard} \\ C_{i,1}^{swap-out} & \hat{p}_{i,1} = \texttt{Swap} \end{cases} \quad (8)$$

With this estimated space-time efficiency, the scheduler initializes request priority $\theta_{i,1}$ with Equation 4.

*(2) Post-Invocation Refinement:* When the external call returns, the request transitions from the tool-wait state to the resumption state. *AugServe* updates the resource consumption based on actual runtime feedback. Specifically, the realized return length and the context-handling policy $p_{i,k-1}$ applied during the previous waiting phase are fed back to refine the cost of the next segment. To capture these evolving resource demands, *AugServe* first computes a feedback realization term $C_{i,k}^{fb}$ that quantifies the actual overhead incurred during the transition, including the context resumption stage ($R_{i,2}^{res}(p_i)$ in Figure 5) and the incremental prefill stage for return tokens ($P_{i,2}^{ret}$ in Figure 5):

$$C_{i,k}^{fb}(p_{i,k-1}) = C_{i,k}^{return} + \underbrace{\begin{cases} 0 & p_{i,k-1} = \texttt{Preserve} \\ C_{i,1}^{recompute} & p_{i,k-1} = \texttt{Discard} \\ C_{i,1}^{swap-in} & p_{i,k-1} = \texttt{Swap} \end{cases}}_{\text{Resumption Cost}}$$
$$(9)$$

Here, Resumption Cost reflects the overhead of restoring the execution state under $p_{i,k-1}$. For instance, if $p_{i,k-1}$ adopts the `Discard` policy, this term captures the KV cache recomputation cost. Return tokens cost $C_{i,k}^{return}$ reflects the memory required to process the call returned tokens. By replacing predictive estimates with realized execution feedback, $C_{i,k}^{fb}$ resolves uncertainty from prior predictions. Subsequently, *AugServe* predicts the remaining execution ($D_{i,2}$ in Figure 5) and any subsequent external calls ($W_{i,2}$ in Figure 5) to construct the final $C_{i,k}$:

$$C_{i,k}(\hat{p}_{i,k}) = C_{i,k}^{fb}(p_{i,k-1}) + C_{i,k}^{decode} + C_{i,k}^{call}(\hat{p}_{i,k}). \quad (10)$$

This segment-level refinement uses realized execution feedback to correct prior predictive estimates, aligning scheduling decisions with actual system states and effectively mitigating inefficiencies from earlier predictions. For requests comprising multiple service segments, this refinement process repeats iteratively until the request completes.

**Scheduling Procedure.** At each scheduling decision, the scheduler proceeds as follows:

- **Cost Construction:** For each request, construct its space-time cost $C_{i,k}$ and compute the priority $\theta_{i,k}$.

- **Ranking:** All requests in the global pool $U^e$ are ranked in descending order of their current value density $\theta_i$.
- **Greedy Packing:** Requests are selected greedily from the ranked queue until the aggregate memory footprint $\sum x_i^e \cdot m_i(t)$ reaches the system memory limit $M^e$.

This procedure dynamically schedules requests based on state-aware space-time costs and value-density priorities, adapting to evolving request states and external calls. The complete pseudocode is provided in Appendix A, and a theoretical analysis is provided in Appendix E.

## 5.4. Dynamic Batch-Level Token Budget

In the scheduling procedure (§5.3), the memory constraint $M^e$ governs greedy packing through a batch-level token budget that limits the number of tokens processed per iteration. However, using a fixed token budget can under-utilize GPU memory or trigger unnecessary discarding of paused request contexts in augmented LLM inference. We dynamically adjust the token budget based on (i) currently available GPU memory $G_{free}(t)$, and (ii) reclaimable memory $G_{kv}^{preempt}(t)$ from preemptable paused requests, i.e., those selecting `Swap` or `Discard` policies by Equation 6. Formally, the token budget $\mathcal{B}_{token}(t)$ is computed as:

$$\mathcal{B}_{token}(t) = \left\lfloor \frac{G_{free}(t) + G_{kv}^{preempt}(t)}{\mu} \right\rfloor, \quad (11)$$

where $\mu$ denotes the per-token memory footprint. To prevent instability from transient memory fluctuations, $\mathcal{B}_{token}(t)$ is clipped to $[\beta_{low} \cdot T_{\max}, \beta_{high} \cdot T_{\max}]$, where $T_{\max}$ is a reference offline budget and $\beta_{low}, \beta_{high}$ are scaling factors. This design balances throughput with controlled use of preemptable paused requests' memory while maintaining scheduling stability under fluctuating GPU memory conditions.

## 6. Evaluation

We evaluate our approach and baselines across different hardware, models, and datasets.

- **Baselines:** We compare *AugServe* against vLLM and INFERCEPT with FCFS scheduling, and a Speculative Shortest-Job-First (SSJF) baseline representing length-based scheduling using predicted output length.
- **Setup and models:** We run GPT-J-6B on an RTX 4090 GPU (24GB), OPT-13B on an H800 GPU (80GB), Vicuna-13B on two A100 GPUs (40GB each), and Llama-3-70B-Instruct on four A100 GPUs.
- **Datasets:** We utilize the INFERCEPT dataset and the ToolBench dataset (Qin et al., 2024b), which features thousands of API calls across various categories.
- **Workloads: (W1)** Poisson arrivals over 30 minutes; **(W2)** Gamma arrivals with varying coefficients of variation.
- **Metrics and SLOs:** We report Time-to-First-Token

(TTFT), normalized latency (end-to-end latency divided by output length), and goodput (number of SLO-satisfying requests per unit time). Following prior work (Gao et al., 2025; Wu et al., 2023), SLOs are set as TTFT<1s and normalized latency<10× single-iteration time. Performance improvements of *AugServe* over baselines are reported as the geometric mean across all experimental settings.

### 6.1. End-to-End Performance

**(1) Goodput.** Figure 7 reports the goodput of *AugServe* and the baselines across three model configurations and request rates under the W1 workload. *AugServe* consistently outperforms all baselines, achieving a geometric mean goodput of $7.5\times$ that of vLLM, $5.7\times$ that of INFERCEPT, and $1.9\times$ that of SSJF. The performance advantage is particularly pronounced under high contention. For example, on the H800 GPU with the ToolBench dataset, at a request rate of 5.0 req/s, the goodput of vLLM and INFERCEPT drops to near zero, whereas *AugServe* sustains a goodput of 2.4 req/s. This gap arises because FCFS scheduling in vLLM and INFERCEPT exacerbates HoL blocking under increasing load. As a result, queuing delays frequently exceed SLOs, sharply reducing SLO attainment (i.e., the fraction of requests meeting SLOs, Appendix D.8) and thereby degrading goodput. SSJF partially alleviates HoL blocking by prioritizing requests based on predicted output lengths. However, it ignores the execution-stage heterogeneity and latency variability introduced by external calls, leading to suboptimal performance under heavy load. In contrast, *AugServe* continuously refines request priorities by incorporating external-call behaviors and cross-round execution states. This state-aware design effectively mitigates HoL blocking in augmented LLM inference, sustaining high SLO attainment under peak request rates (Appendix D.8) and thereby achieving substantially higher goodput.

**(2) Latency Performance.** Figure 7 also compares the TTFT and normalized latency of *AugServe* with the baselines. These two metrics jointly influence goodput, capturing request-level responsiveness and token-level execution efficiency, respectively.

TTFT reflects the queuing delay before the first token is generated. Across all evaluated scenarios, *AugServe* consistently achieves substantially lower TTFT, with geometric mean reductions of 95.6%, 96.0%, and 92.5% compared to vLLM, INFERCEPT, and SSJF, respectively. Under heavy load, existing systems suffer from severe HoL blocking caused by requests stalled at external calls, which prevents the scheduler from prioritizing runnable and resource-efficient executions. In contrast, *AugServe* maintains high responsiveness by scheduling requests based on their execution stage and state-dependent resource efficiency. Meanwhile, normalized latency captures average token-level ex-

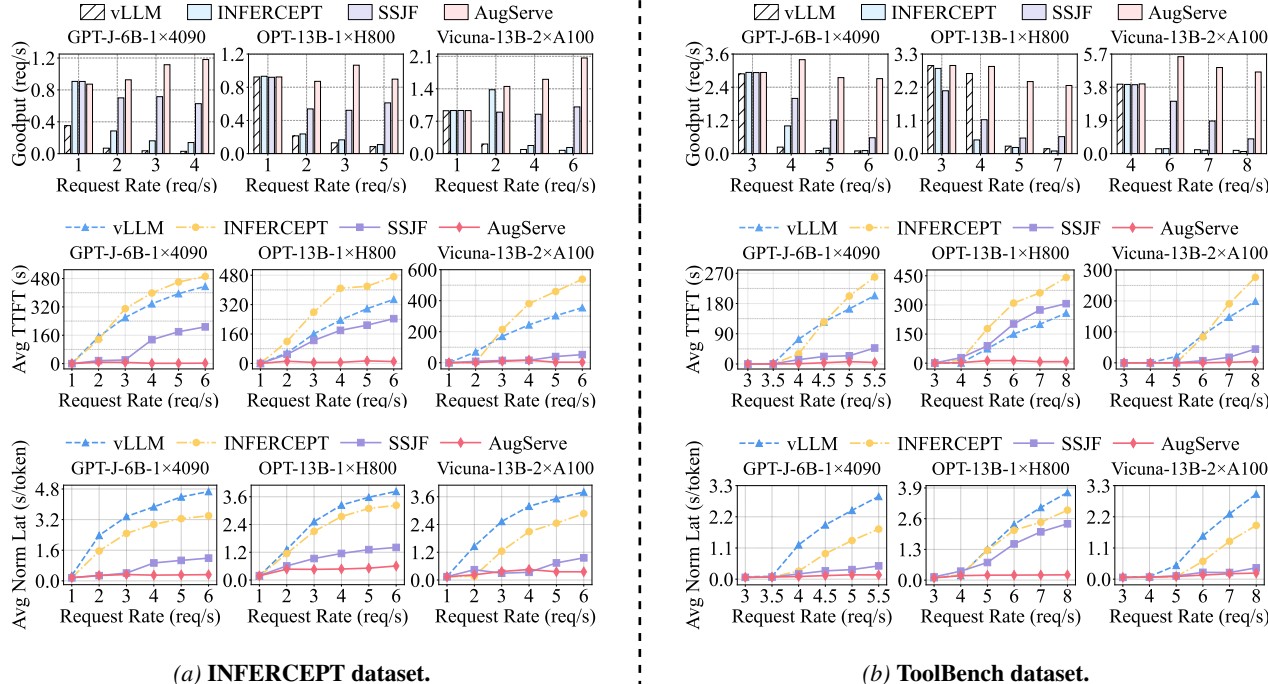

*(a)* **INFERCEPT dataset.**

*(b)* **ToolBench dataset.**

*Figure 7.* Comparison of goodput, TTFT, and normalized latency among vLLM, INFERCEPT, Speculative-SJF scheduling, and AugServe with different models and GPUs on INFERCEPT and ToolBench datasets. First row: Goodput (higher is better). Second row: TTFT (lower is better). Third row: Normalized latency (lower is better).

ecution efficiency while mitigating bias from varying sequence lengths, enabling fair comparison across requests. Overall, *AugServe* reduces normalized latency by 77.8%, 72.8%, and 36.5% relative to vLLM, INFERCEPT, and SSJF, respectively. These gains in both responsiveness and token-level efficiency contribute to *AugServe*'s superior goodput. Additional analyses of tail latency (P95) are provided in Appendix D.7.

**(3) Additional Results.** Appendix D presents additional experimental results that further validate the effectiveness and generality of *AugServe*. We first evaluate *AugServe* under diverse models and workload settings, including mixed external-call and non-external-call workloads (D.1) and Llama-3-70B-Instruct (D.2). Then, we provide complementary analyses of sensitivity to $\beta$ (D.3), robustness to bursty arrivals (D.5), and comparison with memory-based SJF scheduling (D.6). Finally, we analyze the GPU memory occupancy (D.9).

### 6.2. Ablation Study

**(1) Ablation.** We conduct an ablation study to evaluate the contributions of *AugServe*'s two core components: dynamic batch-level token budget and state-aware scheduling. Experiments use OPT-13B on an H800 GPU with a 3.0 req/s workload on the INFERCEPT dataset. Table 2 summarizes their impact on TTFT and goodput, with Base denoting the

*Table 2.* Ablation study of *AugServe*.

|       | TTFT (s) | Goodput (req/s) |
|-------|----------|-----------------|
| Base  | 306.26   | 0.15            |
| +B    | 293.51   | 0.21            |
| +S    | 14.53    | 0.93            |
| Aug   | 3.88     | 1.07            |

INFERCEPT baseline.

We first enable the dynamic batch-level token budget alone (+B in Table 2) on INFERCEPT. By adapting batch capacity to available GPU memory, including reclaimable memory from paused requests, this component improves resource utilization and increases goodput by 41.5%. Next, we replaced the original FCFS algorithm with AugServe's state-aware scheduling (+S in Table 2) while keeping a fixed batch budget. This reduces average TTFT by 95.2% and boosts goodput by $4.2\times$. Finally, combining both components in the full *AugServe* (Aug in Table 2) achieves the best overall performance, yielding lower TTFT and higher goodput than either component alone. This confirms that dynamic batch-level capacity adaptation and state-aware scheduling are complementary and jointly essential for efficient augmented LLM inference.

**(2) Latency Breakdown.** Figure 9 shows the breakdown of end-to-end request latency for *AugServe* and the baselines.

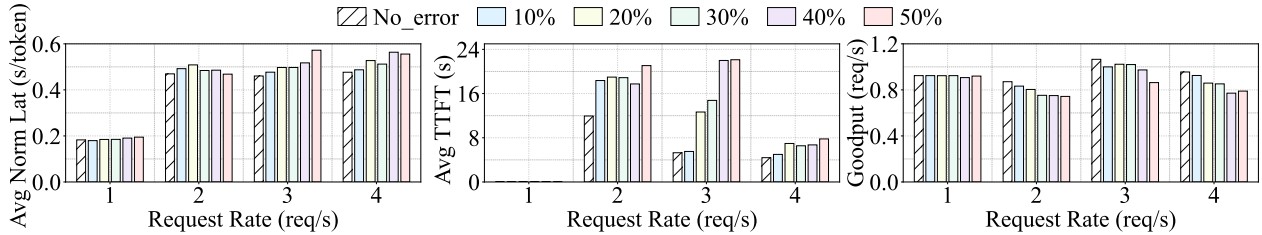

*Figure 8.* Comparison of normalized latency, TTFT, and goodput under prediction error injection with OPT-13B on an H800 GPU using the INFERCEPT dataset. Prediction errors are injected at 10%, 20%, 30%, 40%, and 50%. Left: Normalized latency (lower is better). Middle: TTFT (lower is better). Right: Goodput (higher is better).

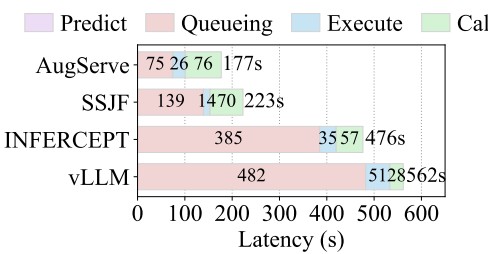

*Figure 9.* Latency breakdown.

The results indicate that *AugServe*'s performance gains are primarily attributed to a substantial reduction in queuing time. Moreover, the prediction module in *AugServe* incurs negligible overhead, accounting for 0.1% of end-to-end latency on average (0.18 s per request).

### 6.3. Robustness to Prediction Errors

To assess the robustness of *AugServe* to prediction inaccuracies, we inject noise into both *external_call_duration* and *output_length* predictions. Specifically, for each prediction $\hat{m}$, we use $\hat{m}(1 + \epsilon)$ as the noisy prediction, where $\epsilon$ is randomly chosen from $\{-p, +p\}$ with equal probability. We vary $p$ from 10% to 50% and evaluate OPT-13B on an H800 GPU using the INFERCEPT dataset.

Figure 8 reports the impact of prediction errors on normalized latency, TTFT, and goodput. Across all error levels, goodput and normalized latency remain largely stable, showing that *AugServe* is insensitive to moderate-to-large prediction noise. TTFT increases mildly as the prediction error grows, with an absolute increase of roughly 10–20 s even under high error rates of 40–50%. Overall, the performance impact of mispredictions remains limited, demonstrating that *AugServe* is robust to prediction inaccuracies. This robustness arises from *AugServe*'s feedback-driven design, which continuously refines scheduling decisions using realized runtime information upon external call returns, rather than relying solely on static predictions.

*Table 3.* Average per-iteration scheduling overhead (s) under different request rates on the INFERCEPT dataset with OPT-13B and an H800 GPU.

| Req/s | INFERCEPT | *AugServe* |
|-------|-----------|-----------|
| 3.0 | 0.013 | 0.018 |
| 4.0 | 0.015 | 0.022 |

### 6.4. Scheduling Overhead

Since *AugServe* performs iteration-level scheduling, we further measure its per-iteration scheduling overhead. Table 3 reports the average scheduling overhead on the INFERCEPT dataset with OPT-13B on an H800 GPU. *AugServe* incurs slightly higher overhead than INFERCEPT due to its richer state-aware scheduling logic, but the overhead remains small. Compared with the substantial queuing-time reduction achieved by *AugServe*, this additional overhead is negligible. Additional scalability results under different numbers of paused requests are provided in Appendix D.4.

## 7. Conclusion

This paper presents *AugServe*, a system for improving inference efficiency in augmented LLM services. *AugServe* combines adaptively state-aware request scheduling with a dynamic batch-level token budget to reduce queuing delays and substantially improve effective throughput.

## Acknowledgements

We would like to thank the anonymous reviewers for their tremendous feedback and comments, which have substantially improved the content and presentation of this paper.

## Impact Statement

This paper presents work whose goal is to advance the field of machine learning systems. There are many potential societal consequences of our work, none of which we feel must be specifically highlighted here.

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

*Table 4.* Notation and definitions of key symbols.

| Symbol | Definition |
|---|---|
| $R_i$ | The $i$-th request in the system |
| $n_i$ | Total number of service segments for request $R_i$ |
| $S_{i,k}$ | The $k$-th service segment of request $R_i$. |
| $P_{i,1}$ | Standard prompt prefill stage in the initial segment. |
| $D_{i,k}$ | Decoding stage within segment $k$. |
| $W_{i,k}$ | Tool-wait stage where the request awaits external responses. |
| $R_{i,k}^{res}(p_i)$ | Context resumption stage, governed by policy $p_i$. |
| $P_{i,k}^{ret}$ | Incremental prefill for processing returned tool tokens. |
| $\Pi$ | Set of context-handling policies: {`Preserve`, `Swap`, `Discard`}. |
| $V_i^e$ | Scheduling value for request $i$ at iteration $e$. |
| $w_i^e$ | Cumulative waiting time for request $i$ up to iteration $e$. |
| $\beta$ | Starvation prevention aging factor. |
| $x_i^e$ | Binary decision variable: 1 if request $i$ is scheduled, 0 otherwise. |
| $M^e$ | Instantaneous GPU memory capacity constraint at iteration $e$. |
| $m_i(t)$ | Instantaneous memory occupancy of request $i$ at time $t$. |
| $C_{i,k}(p_{i,k})$ | Space-time cost (action) of segment $S_{i,k}$ under policy $p_{i,k}$. |
| $\tau_{i,k}$ | Total residency duration of the request segment. |
| $\theta_{i,k}$ | Scheduling priority defined as the value density $V_i/C_{i,k}$. |
| $\hat{l}_i^{out}$ | Predicted output token length. |
| $\hat{\tau}_i^{call}$ | Predicted duration of the external tool call. |
| $\hat{p}_{i,k}$ | Predicted context policy during the wait stage. |
| $C_{i,k}^{fb}$ | Feedback-driven realized cost term for $k > 1$. |
| $p_{i,k-1}$ | Context policy actually executed during the previous wait phase. |
| $\mathcal{B}_{token}(t)$ | Dynamic batch-level token budget at time $t$. |
| $G_{free}(t)$ | Number of currently available free GPU memory blocks. |
| $G_{kv}^{preempt}(t)$ | Reclaimable memory blocks held by paused request contexts. |
| $\mu$ | Memory footprint per token. |
| $T_{max}$ | Static offline reference token budget. |
| $\beta_{low}, \beta_{high}$ | Scaling factors for the smoothing range of the token budget. |

## A. State-Aware Adaptive Scheduling Algorithm

Table 4 summarizes the symbols used throughout the paper. The overall workflow of *AugServe* is presented in algorithm 1 , which integrates state-aware adaptive scheduling with dynamic batch-level token budget.

Upon request arrival, *AugServe* invokes a lightweight prediction module to estimate the *external_call_duration* and *output_length*. These estimates are used to select an initial context-handling policy and to compute the request's initial scheduling priority based on its expected execution characteristics (Lines 4–10).

When an external tool call completes, *AugServe* performs state-aware refinement by replacing predictive estimates with realized runtime feedback. The scheduling cost and priority are updated based on the actual context-handling policy and return length, and the request is reinserted into the appropriate queue according to its execution state (Lines 12–25).

To respect runtime memory constraints, *AugServe* dynamically adjusts the batch-level token budget based on available GPU memory and reclaimable memory from paused requests under different context-handling policies. Bounded smoothing is applied to prevent abrupt budget fluctuations and ensure system stability (Lines 27–28).

Finally, *AugServe* globally ranks all runnable requests according to their refined scheduling priorities (Line 30). A batch is then greedily constructed under the current token budget constraint (Lines 32–40) and executed in the next forward iteration.

---

**Algorithm 1** *AugServe*: State-Aware Adaptive Scheduling with Dynamic Token Budget

---

1: **Input:** Request queues: $\text{running}, \text{swapped}, \text{waiting}, \text{paused}$; Predictor Pred; Aging parameter $\beta$; Scaling bounds $\beta_{low}, \beta_{high}$.

2: **while** True **do**

3:     `// (1) Request arrival & Predictive Initialization`

4:     **for** each $r \in \text{arrivals}$ **do**

5:         $(\hat{\tau}_i^{call}, \hat{l}_i^{out}) \leftarrow \text{Pred}(r)$

6:         $\hat{p}_{i,1} \leftarrow \arg\min_{p \in \Pi} \text{Waste}_i^p(\hat{l}_i^{out}, \hat{\tau}_i^{call})$     // Select policy

7:         Estimate $\hat{C}_{i,1}$ using $\hat{p}_{i,1}$ and $(\hat{\tau}_i^{call}, \hat{l}_i^{out})$ by Equation 5

8:         $r.\theta_{i,1} \leftarrow (1 + \beta \cdot w_i)/\hat{C}_{i,1}$ by Equation 2 and Equation 4

9:         $\text{waiting.push}(r)$

10:    **end for**

11:    `// (2) State-Aware refinement upon tool return`

12:    **for** each $(r, p_{i,k-1}) \in \text{paused}$ **do**

13:        **if** $r.\text{apiFinished}()$ **then**

14:           Update $C_{i,k}^{fb}$ using realized policy $p_{i,k-1}$ by Equation 9

15:           $C_{i,k} \leftarrow C_{i,k}^{fb} + C_{i,k}^{decode} + \hat{C}_{i,k}^{call}$

16:           **if** $p_{i,k-1} = \text{Preserve}$ **then**

17:              $\text{running.push}(r)$

18:           **else if** $p_{i,k-1} = \text{Swap}$ **then**

19:              $\text{swapped.push}(r)$

20:           **else**

21:              $\text{waiting.push}(r)$     // Discard policy

22:           **end if**

23:           Update priority $r.\theta_{i,k} \leftarrow (1 + \beta \cdot w_i)/C_{i,k}$

24:        **end if**

25:    **end for**

26:    `// (3) Dynamic token budget adjustment`

27:    $\mathcal{B}_{token}(t) \leftarrow \lfloor (G_{free}(t) + G_{kv}^{preempt}(t))/\mu \rfloor$ by Equation 11

28:    $\mathcal{B}_{token}(t) \leftarrow \text{clip}(\mathcal{B}_{token}(t), \beta_{low} \cdot T_{max}, \beta_{high} \cdot T_{max})$

29:    `// (4) Global queue ranking & Greedy packing`

30:    Sort all $r$ in candidate queues by $\theta_{i,k}$

31:    $\text{scheduled} \leftarrow \emptyset$

32:    **for** each $r \in \text{sorted candidates}$ **do**

33:        **if** $\text{active\_tokens} + \text{tokens}(r) \leq \mathcal{B}_{token}(t)$ **then**

34:           $\text{scheduled} \leftarrow \text{scheduled} \cup \{r\}$

35:           $\text{active\_tokens} \leftarrow \text{active\_tokens} + \text{tokens}(r)$

36:           Update $r.waiting\_time$

37:        **else**

38:           **break**     // Budget reached

39:        **end if**

40:    **end for**

41:    `// (5) Execute one iteration`

42:    **forward**(scheduled)

43: **end while**

---

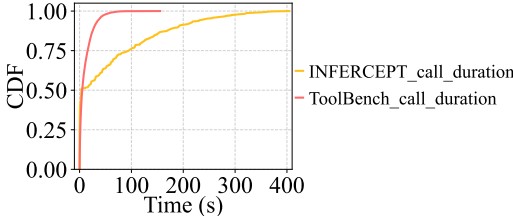

*Figure 10.* External call duration distribution in INFERCEPT and ToolBench datasets.

## B. Context-Handling Policy

The construction of the predictive space-time cost $\hat{C}_{i,k}$ (§5.3) is inherently tied to the choice of context-handling policy $\hat{\pi}_{i,k}$. During an external tool call, the resource consumption of a request depends critically on how its KV cache is managed, reflecting a fundamental trade-off between GPU memory occupancy and computational overhead. Specifically, the `Preserve` policy avoids recomputation by keeping the context in GPU memory throughout the call, but incurs high memory occupancy, whereas the `Discard` policy releases GPU memory at the cost of recomputation when the request resumes. As shown in Figure 3 and Figure 10, external call durations and cumulative context lengths exhibit highly heterogeneous distributions across both datasets. This variability implies that a fixed context-handling policy is suboptimal: short external calls with small context favor preserving context to avoid recomputation overhead, while long calls with large context benefit from releasing GPU memory via discarding or swapping. To minimize the total resource overhead induced by external calls, *AugServe* selects the most efficient context-handling policy based on the request's current context length $L_i$, its predicted *output_length* $\hat{l}_i^{out}$ and predicted *external_call_duration* $\hat{\tau}_i^{call}$ (§5.2). To formalize this optimization, we adopt the memory-waste formulation introduced by INFERCEPT (Abhyankar et al., 2024), where the waste metric captures the opportunity cost of GPU memory being occupied or reclaimed during the external call.

For a request $i$, let $M$ denote the per-token memory footprint, let $\tilde{L}_i = L_i + \hat{l}_i^{out}$ denote the estimated context length at the moment the external call is issued, where $L_i$ is the current context length and $\hat{l}_i^{out}$ is the predicted number of tokens generated before the call. We denote by $\hat{\tau}_i^{call}$ the predicted duration of the external call, and by $\tau^{fwd}(L)$ the execution time of a forward iteration with context length $L$. We further denote by $L_{other}$ the aggregate context length of other runnable requests, $\tau^{swap}(L)$ the time to swap a context of length $L$, and $N_{\max}^{fwd}$ the maximum number of tokens that can be swapped per forward iteration.

**Preserve.** Under the `Preserve` policy, the request retains its context in GPU memory throughout the external call. The resulting memory waste equals the memory footprint multiplied by the call duration:

$$\text{Waste}_i^{\text{Preserve}} = \hat{\tau}_i^{call} \cdot \tilde{L}_i \cdot M. \tag{12}$$

**Discard.** With the `Discard` policy, the context is freed during waiting and recomputed upon resumption. The waste arises from recomputation overhead, including both the request itself and interference with other active requests:

$$\text{Waste}_i^{\text{Discard}} = \tau^{fwd}(\tilde{L}_i) \cdot \tilde{L}_i \cdot M + \tau^{fwd}(\tilde{L}_i) \cdot L_{other} \cdot M. \tag{13}$$

**Swap.** Under the `Swap` policy, the context is temporarily swapped to secondary storage and restored upon resumption. The waste is dominated by bidirectional swapping overhead that may stall concurrent forward execution:

$$\text{Waste}_i^{\text{Swap}} = 2 \cdot \tau^{swap}(\tilde{L}_i) \cdot N_{\max}^{fwd} \cdot M. \tag{14}$$

**Policy Selection.** We select the context-handling policy $\hat{p}_i$ that minimizes the expected memory waste during the external call:

$$\hat{p}_i = \arg \min_{p \in \{\texttt{Preserve},\texttt{Discard},\texttt{Swap}\}} \text{Waste}_i^p. \tag{15}$$

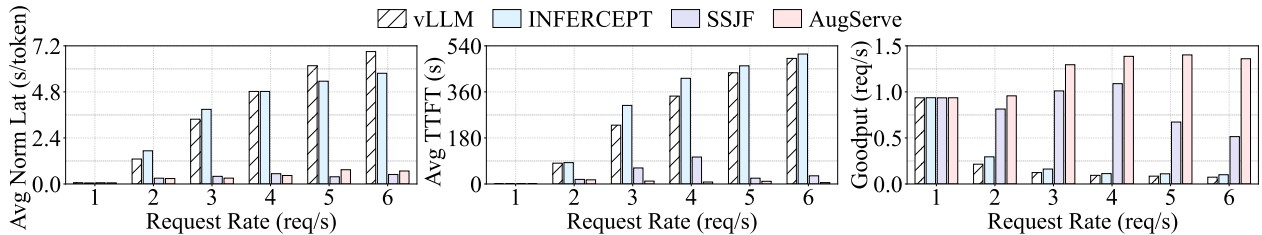

*Figure 11.* Average normalized latency (s/token), TTFT (s), goodput (req/s) comparison among vLLM, INFERCEPT, Speculative-SJF scheduling, and *AugServe* with OPT-13B using mixed workloads of external-call and non-external-call requests on an H800 GPU.

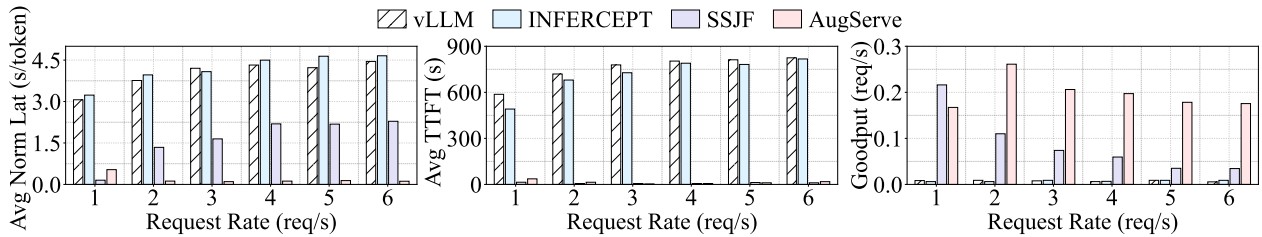

*Figure 12.* Average normalized latency (s/token), TTFT (s), goodput (req/s) comparison among vLLM, INFERCEPT, Speculative-SJF scheduling, and *AugServe* with Llama-3-70B-Instruct using INFERCEPT dataset on 4 A100 GPUs.

While INFERCEPT utilizes these formulas within a First-Come-First-Served (FCFS) scheduling framework, *AugServe* integrates this waste-aware selection into our state-aware cost model. By quantifying waiting-phase overheads in advance, the scheduler can internalize the downstream memory impact of external calls into its value-density computation. This enables state-aware prioritization that aligns immediate execution decisions with long-term global resource efficiency.

## C. Related Work

**Memory Optimizations.** vLLM (Kwon et al., 2023) improves GPU memory utilization with PagedAttention, allocating KV cache in fixed-size blocks to reduce fragmentation. INFERCEPT (Abhyankar et al., 2024) proposes dynamic context-handling policies for augmented LLM inference, selectively preserving, swapping, or discarding KV cache during external-call pauses. Sarathi (Agrawal et al., 2024) employs chunked prefill to interleave prefill and decoding for improved resource utilization. FlexGen (Sheng et al., 2023) and FlashGen (Jeong & Ahn, 2025) further optimize KV cache management by offloading data to CPU memory or SSDs. These techniques primarily focus on memory efficiency and serve as complementary building blocks for high-performance LLM serving.

**Scheduling.** Recent research has explored improving request scheduling for LLM inference. Orca (Yu et al., 2022) leverages iteration-level batching to increase GPU utilization, while FastServe (Wu et al., 2023) employs input-length–aware, token-level preemptive scheduling to mitigate HoL blocking. Other approaches (Jin et al., 2023; Fu et al., 2024; Hu et al., 2025a; Zheng et al., 2023; Shahout et al., 2025a) approximate Shortest-Job-First (SJF) scheduling by predicting output lengths to reduce queuing delay. While effective for text-only inference, these methods do not explicitly account for the heterogeneous and multi-stage execution behavior introduced by external calls in augmented LLM inference. MARS (Shahout et al., 2025b) approximates SJF scheduling by predicting memory usage and making scheduling decisions independently at each execution round. Requests are treated as newly arriving jobs after external calls, without explicitly incorporating realized runtime feedback from prior execution stages. As a result, it lacks an explicit coupling across rounds to model cross-round execution dynamics or the resumption cost induced by cumulative context growth and variability in external call returns. In contrast, *AugServe* introduces a state-aware adaptive scheduling strategy that continuously refines scheduling priorities using runtime feedback upon external call returns. By explicitly capturing execution state transitions and realized resumption costs, *AugServe* corrects predictive inaccuracies and adapts to dynamic external environments, enabling more robust scheduling and higher effective throughput in augmented LLM inference.

*Table 5.* Sensitivity to the balancing parameter $\beta$ on the INFERCEPT dataset with OPT-13B on an H800 GPU. The default value used in the main experiments is marked with *.

| Metric | Req/s | $\beta = 10^{-3}$ | $\beta = 10^{-4}$ | $\beta = 5\times10^{-5}$ * | $\beta = 10^{-5}$ | $\beta = 10^{-6}$ |
|---|---|---|---|---|---|---|
| Goodput (req/s) | 3.0 | 1.01 | 1.12 | 1.07 | 1.04 | 1.02 |
| | 5.0 | 0.82 | 0.94 | 0.90 | 0.94 | 0.86 |
| P99 TTFT (s) | 3.0 | 276.02 | 109.05 | 98.66 | 111.89 | 278.10 |
| | 5.0 | 298.65 | 152.84 | 157.45 | 166.99 | 226.58 |

## D. Additional Experimental Results

### D.1. Mixed Workloads of External-Call and non-External-Call Requests

Figure 11 reports the average normalized latency, TTFT, and goodput for vLLM, INFERCEPT, SSJF, and *AugServe* under mixed workloads of external-call and non-external-call requests. We construct the mixed workload from the INFERCEPT dataset by disabling external calls for 50% of the requests, while preserving the original prompts and output lengths. All experiments are conducted using OPT-13B on an H800 GPU. The results show that *AugServe* consistently achieves the best performance across all metrics. Under mixed workloads, FCFS-based systems suffer from severe HoL blocking when external-call requests stall execution, which propagates queuing delays to non-external-call requests and degrades overall responsiveness. SSJF partially mitigates this issue by prioritizing shorter requests, but remains unaware of external-call-induced execution states and thus yields limited gains. In contrast, *AugServe* explicitly accounts for heterogeneous execution states across requests, effectively mitigating the blocking impact of stalled external-call requests on fast non-external-call requests and maintaining high goodput under mixed workloads.

### D.2. Llama-3-70B-Instruct Results

Figure 12 compares average normalized latency, TTFT, and goodput across vLLM, INFERCEPT, SSJF scheduling, and *AugServe* when serving Llama-3-70B-Instruct on four A100 GPUs using the INFERCEPT dataset. Despite the significantly larger model size and KV cache footprint, *AugServe* consistently outperforms all baselines across all metrics. FCFS-based systems suffer from severe queuing delays under external-call-augmented workloads, while SSJF provides only limited improvement due to its lack of awareness of external-call-induced execution states and cross-round resumption dynamics. In contrast, *AugServe* maintains low TTFT and high goodput, demonstrating that its state-aware scheduling and dynamic batch-level adaptation remain effective under large-model, multi-GPU inference serving.

### D.3. $\beta$ Sensitivity

We further evaluate the sensitivity of *AugServe* to the balancing parameter $\beta$ in Equation 2, which balances throughput progress and waiting-time reduction in the scheduling value. All experiments are conducted on the INFERCEPT dataset with OPT-13B on an H800 GPU. The default value used in the main experiments is $\beta = 5 \times 10^{-5}$. As shown in Table 5, *AugServe* is not overly sensitive to the choice of $\beta$. Across different request rates, strong performance is consistently achieved within a stable range around $10^{-5}$–$10^{-4}$, rather than at a single sharply tuned value. The default value $\beta = 5 \times 10^{-5}$ falls within this stable range and achieves balanced performance across workloads. When $\beta$ is too large, e.g., $\beta = 10^{-3}$, the scheduler over-emphasizes waiting time and may prioritize long-waiting requests at the cost of execution efficiency, leading to worse tail latency and lower goodput. When $\beta$ is too small, e.g., $\beta = 10^{-6}$, the scheduler behaves closer to a throughput-dominant greedy policy, which weakens delay awareness and can degrade tail latency. Overall, *AugServe* remains effective over a relatively wide range of $\beta$ values and does not require fine-grained tuning for specific workloads.

### D.4. Scheduling Overhead with Increasing Paused Requests

In §6.4, we report the average per-iteration scheduling overhead of *AugServe* under different request rates. Here, we further examine how this overhead changes as the number of paused requests increases, which is important for augmented LLM workloads where many requests may concurrently wait for external call returns. As shown in Table 6, the per-iteration scheduling overhead of *AugServe* increases slightly with the number of paused requests, but remains at the millisecond

*Table 6.* Average per-iteration scheduling overhead (s) under different numbers of paused requests.

| Paused Requests | INFERCEPT | *AugServe* |
|:---:|:---:|:---:|
| ∼50 | 0.002 | 0.002 |
| 50–100 | 0.014 | 0.008 |
| 100–150 | 0.013 | 0.016 |
| 150–200 | – | 0.025 |
| ≥200 | – | 0.021 |

*Table 7.* Goodput (req/s) comparison of vLLM, INFERCEPT, Speculative-SJF scheduling, and *AugServe* under different request rates and arrival burstiness (CV) on INFERCEPT dataset with OPT-13B and an H800 GPU.

| Request Rate (req/s) | CV | vLLM | INFERCEPT | SSJF | *AugServe* |
|:---:|:---:|:---:|:---:|:---:|:---:|
| 2.0 | 1 | 0.21 | 0.28 | 0.49 | 0.85 |
| | 1.5 | 0.09 | 0.13 | 0.52 | 0.82 |
| | 2 | 0.06 | 0.08 | 0.39 | 0.78 |
| 3.0 | 1 | 0.14 | 0.20 | 0.50 | 1.03 |
| | 1.5 | 0.07 | 0.11 | 0.54 | 0.81 |
| | 2 | 0.04 | 0.06 | 0.23 | 0.42 |

level even with hundreds of paused requests. This indicates that iteration-level scheduling does not become a bottleneck under high concurrency. This behavior is expected, as the scheduling procedure is mainly dominated by sorting and greedy selection over candidate requests, whose cost scales smoothly with the number of active and paused requests. Overall, *AugServe* maintains stable and efficient scheduling under high concurrency.

### D.5. Robustness to Bursty Arrivals

To evaluate robustness under bursty traffic, we model request arrivals using a Gamma distribution and control burstiness via the coefficient of variation (CV). We fix the average request rate and vary CV to induce different levels of load fluctuation, using the INFERCEPT dataset with OPT-13B on an H800 GPU. Table 7 reports goodput under different burst levels. Across all evaluated load conditions, *AugServe* consistently exhibits more stable performance than baselines. As burstiness increases, the goodput of FCFS-based systems degrades sharply, while SSJF offers only limited improvement due to its lack of awareness of external-call-induced execution states. At 2.0 req/s with CV = 1.5, the goodput of vLLM and INFERCEPT drops to approximately 0.1 req/s, whereas *AugServe* sustains a throughput of 0.82 req/s. This result demonstrates that *AugServe* is significantly more resilient to bursty arrivals, benefiting from its state-aware scheduling and adaptive capacity control that mitigate burst-induced queue buildup.

### D.6. Comparison with Memory-Based SJF Scheduling

We further compare *AugServe* with MARS, a representative memory-based SJF scheduler for augmented LLM inference, which prioritizes requests based on estimated memory cost. Following MARS, each service round is treated as an independent scheduling unit with static, per-call cost estimation, without explicitly modeling cross-round execution state. Since MARS does not provide a prediction model for the INFERCEPT workload, we re-implement its scheduling policy and priority formulation within our system and evaluate it under the same prediction interface as other baselines to ensure a fair comparison. Consistent with prior observations, memory-based SJF improves over FCFS-style baselines under light to moderate load, reducing TTFT and normalized latency while maintaining higher goodput. However, as load increases and requests exhibit larger cumulative context growth and higher variability in external call returns, its per-round static cost abstraction becomes increasingly brittle, leading to degraded performance under heavy load. In some scenarios, its performance can approach or even fall below simpler length-based SJF heuristics. In contrast, *AugServe* explicitly models execution-stage-dependent state and dynamically refines scheduling decisions across service rounds, enabling more stable performance across load levels. As shown in Figure 13 and Figure 14, *AugServe* achieves a geometric mean goodput of $3.07\times$ that of MARS across the INFERCEPT and mixed datasets, and up to $4.78\times$ higher under heavy load. It also reduces TTFT by 92.1% and normalized latency by 38.9%, demonstrating both higher efficiency and robustness.

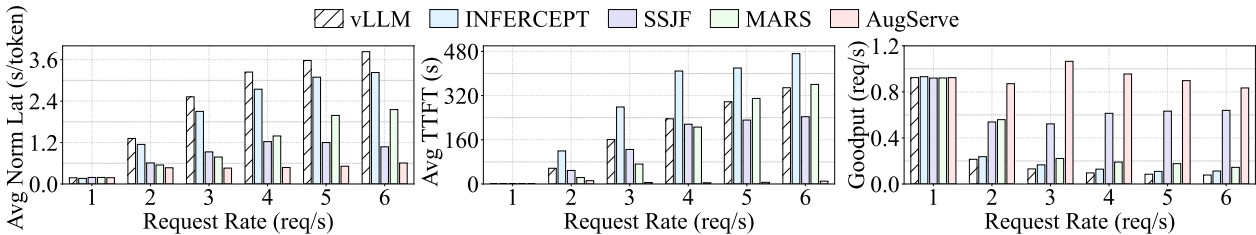

*Figure 13.* Average normalized latency (s/token), TTFT (s), goodput (req/s) comparison among vLLM, INFERCEPT, Speculative-SJF scheduling, MARS, and *AugServe* with OPT-13B using INFERCEPT dataset on an H800 GPU.

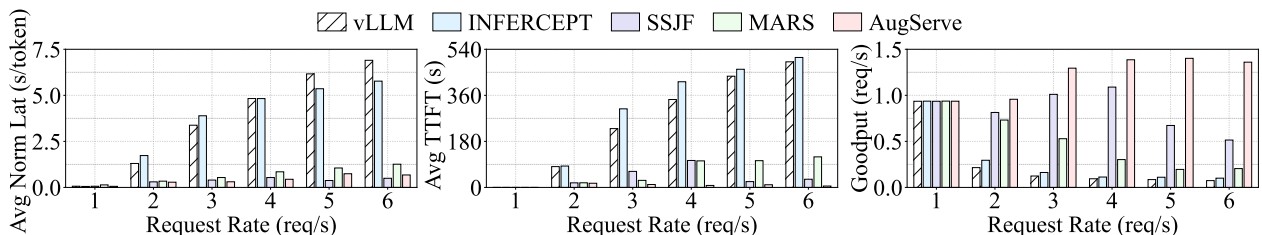

*Figure 14.* Average normalized latency (s/token), TTFT (s), goodput (req/s) comparison among vLLM, INFERCEPT, Speculative-SJF scheduling, MARS, and *AugServe* with OPT-13B using mixed workloads of external-call and non-external-call requests on an H800 GPU.

### D.7. Tail Latency Performance

Figure 15 and Figure 16 report the P95 TTFT and P95 normalized latency under different load levels, using the same experimental setup as in §6.1. As load increases, the tail latency of vLLM and INFERCEPT grows sharply, reflecting severe queuing delays and resource contention under high load. SSJF provides limited improvement by prioritizing shorter requests, but still exhibits high tail latency when queuing delay is dominated by external-call-induced stalls. In contrast, *AugServe* consistently maintains low P95 TTFT and normalized latency across load levels, demonstrating robust tail latency control under external-call-augmented workloads.

### D.8. SLO Attainment

To better understand the source of goodput gains, we report SLO attainment under the same experimental settings as in §6.1. SLO attainment measures the fraction of requests that satisfy latency constraints and directly determines goodput. As shown in Figure 17, *AugServe* consistently achieves higher SLO attainment than all baselines, especially under higher load levels. In contrast, FCFS-based systems quickly violate SLOs as queuing delays grow, while SSJF provides only limited improvement due to its lack of awareness of external-call-induced execution states. These results explain the goodput improvements observed in §6.1, confirming that *AugServe* sustains higher goodput by maintaining high SLO attainment under contention.

### D.9. Memory Occupancy

Figure 18 illustrates the GPU and CPU cache occupation across inference iterations. SSJF maintains low GPU utilization, with cache occupancy remaining below 50% in most iterations and negligible CPU offloading. This conservative behavior avoids memory pressure but leads to underutilized GPU resources and limited throughput. vLLM rapidly saturates GPU memory without CPU offloading, indicating a lack of mechanisms to manage paused requests under external calls. As a result, memory pressure accumulates on GPU, exacerbating queuing delays and HoL blocking. INFERCEPT actively offloads KV cache to CPU to relieve GPU pressure, but its CPU cache occupation increases steadily and remains high in later iterations, reflecting frequent context swapping and significant offloading overhead. Aggressive CPU offloading improves memory availability but introduces substantial resumption overhead, which is particularly harmful under frequent external-call-induced pauses. *AugServe* achieves high GPU utilization while maintaining moderate and stable CPU cache

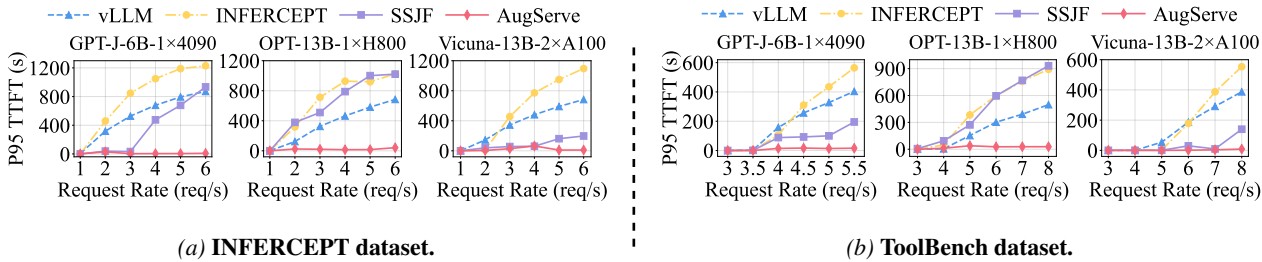

*(a)* **INFERCEPT dataset.**      *(b)* **ToolBench dataset.**

*Figure 15.* P95 Time-to-First-Token (TTFT) (s) comparison among vLLM, INFERCEPT, Speculative-SJF scheduling, and *AugServe* on INFERCEPT and ToolBench datasets with different models and GPUs. Lower right is better, i.e., shorter response time and queuing time.

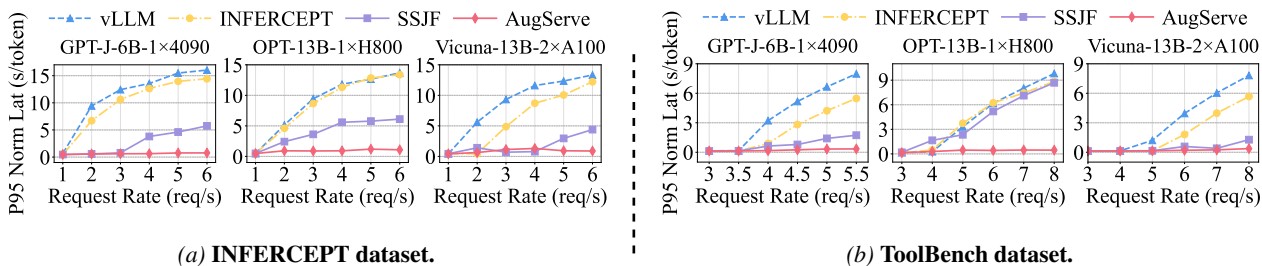

*(a)* **INFERCEPT dataset.**      *(b)* **ToolBench dataset.**

*Figure 16.* P95 normalized Latency (s/token) comparison among vLLM, INFERCEPT, Speculative-SJF scheduling, and *AugServe* on INFERCEPT and ToolBench datasets with different models and GPUs. Lower right is better, i.e., sustains higher serving load.

occupation. By explicitly accounting for reclaimable memory and dynamically adapting batch capacity, *AugServe* controls memory pressure and avoids excessive context eviction and CPU offloading, leading to more balanced resource utilization and stable performance.

## E. Approximation Guarantee

This section analyzes the value-density greedy policy under a simplified per-iteration scheduling formulation. Consider a fixed candidate set $U$, fixed scheduling values $v_i > 0$, fixed memory costs $c_i > 0$, and a fixed memory budget $B$. The per-iteration scheduling problem is:

$$\max \sum_{i \in U} v_i x_i \quad \text{s.t.} \quad \sum_{i \in U} c_i x_i \leq B, \quad x_i \in \{0, 1\}. \tag{16}$$

This is a standard 0–1 knapsack problem. We discard infeasible singleton actions with $c_i > B$, since each candidate action is treated as indivisible in this simplified 0–1 formulation and thus cannot appear in any feasible solution. Let $\rho_i = v_i/c_i$ denote the value density, and sort requests such that $\rho_1 \geq \rho_2 \geq \cdots \geq \rho_n$. Let $G$ be the feasible set obtained by density-based greedy packing, which scans requests in this order and includes a request whenever it fits. Let $S$ be the best feasible singleton:

$$S = \arg \max_{i : c_i \leq B} v_i. \tag{17}$$

The algorithm returns $\text{ALG} = \max\{v(G), v(S)\}$.

**Theorem E.1.** *For the simplified per-iteration problem in Equation 16, the algorithm that returns the better of density-based greedy packing and the best feasible singleton achieves a $1/2$-approximation to the optimal 0–1 knapsack solution.*

*Proof.* Let OPT be the optimal integral value. Let $\text{OPT}_{\text{frac}}$ be the optimal value of the fractional relaxation, where each request can be partially selected, i.e., the binary decision $x_i \in \{0, 1\}$ is relaxed to $x_i \in [0, 1]$. Clearly, $\text{OPT} \leq \text{OPT}_{\text{frac}}$. Since requests are sorted by non-increasing value density, the optimal fractional solution takes a density-ordered prefix and possibly a fraction of one additional request. That is, for some index $t$, requests $1, \ldots, t-1$ are fully selected and request $t$

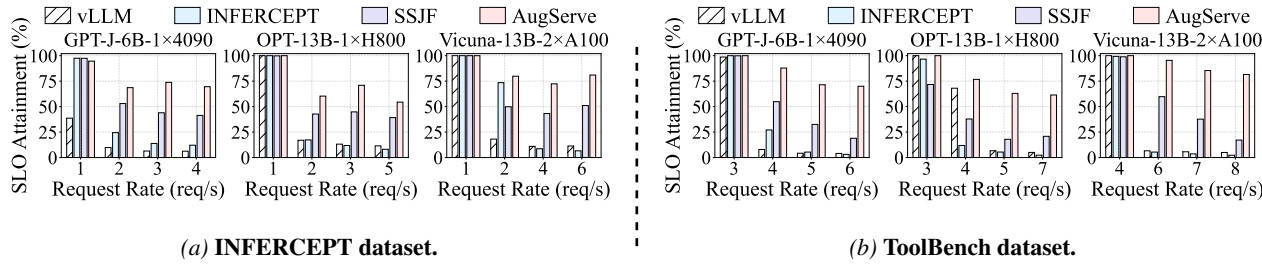

*(a)* **INFERCEPT dataset.**            *(b)* **ToolBench dataset.**

*Figure 17.* SLO attainment (%) with SLOs comparison among vLLM, INFERCEPT, Speculative-SJF scheduling, and *AugServe* on INFERCEPT and ToolBench datasets with different models and GPUs. Higher is better.

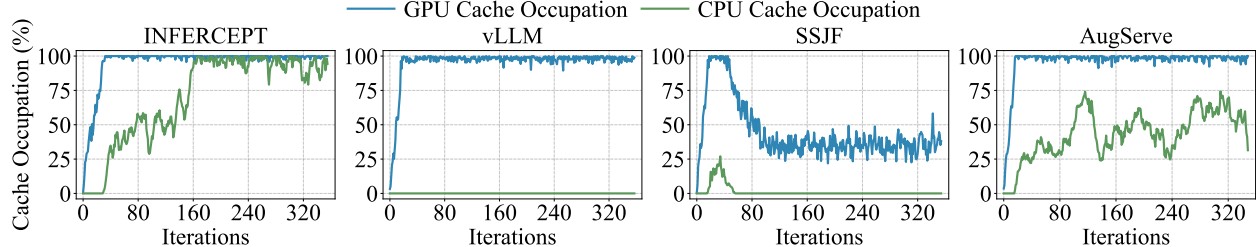

*Figure 18.* GPU/CPU cache occupation comparison among vLLM, INFERCEPT, Speculative-SJF scheduling, and *AugServe* with OPT-13B on an H800 GPU using INFERCEPT dataset.

is partially selected, with

$$\sum_{i=1}^{t-1} c_i \le B \le \sum_{i=1}^{t} c_i. \tag{18}$$

Therefore,

$$\text{OPT}_{\text{frac}} \le \sum_{i=1}^{t-1} v_i + v_t. \tag{19}$$

By construction, the greedy packing solution $G$ includes all requests $1, \ldots, t-1$, because their total cost is at most $B$ and the greedy algorithm scans them before request $t$. Thus, $v(G) \ge \sum_{i=1}^{t-1} v_i$. Moreover, since request $t$ is feasible as a singleton after infeasible requests are removed, the best singleton satisfies $v(S) \ge v_t$. Combining these bounds with [Equation 19](#), we have

$$\text{OPT} \le \text{OPT}_{\text{frac}} \le v(G) + v(S). \tag{20}$$

Finally,

$$\text{ALG} = \max\{v(G), v(S)\} \ge \frac{v(G) + v(S)}{2} \ge \frac{\text{OPT}}{2}. \tag{21}$$

Hence, the algorithm achieves a $1/2$-approximation.      □

**Discussion.** This guarantee applies only to the simplified per-iteration subproblem with fixed values, fixed costs, and a fixed memory budget. It characterizes the greedy packing step in isolation and does not directly extend to the full augmented LLM scheduling problem, where costs evolve across service segments, memory usage depends on external call returns, context-handling policies introduce heterogeneous resumption costs, and scheduling decisions are continuously refined using runtime feedback. Extending formal approximation or competitive guarantees to the full online, multi-stage augmented LLM serving problem remains an important direction for future work.

