# OpenReview forum: "AugServe: Adaptive Request Scheduling for Augmented Large Language Model Inference Serving"
_ICML.cc/2026/Conference — ICML 2026 regular_

### Official Review · Reviewer_pi6b · 2026-03-04

**Soundness:** 3
**Presentation:** 3
**Significance:** 3
**Originality:** 2
**Overall Recommendation:** 5
**Confidence:** 4

**Summary:**

AugServe is a scheduling framework for tool-augmented LLM inference that jointly optimizes request ordering and batch capacity. The central abstraction is a "space-time cost model" that accounts for memory occupancy over time across execution phases (prefill, decode, API wait, swap-in/out, recompute, post-API processing). Requests are scheduled by value density (priority-to-cost ratio) with anti-starvation aging. A two-stage design is used: Stage I (pre-invocation) predicts output length and call duration via a BERT-based model; Stage II (post-invocation) corrects predictions using observed values. A dynamic token-budget module adapts batch capacity based on free GPU memory plus reclaimable memory from paused requests. Experiments across 4 LLMs (GPT-J-6B to Llama-3-70B), 3 GPU types (RTX 4090, H800, A100), and 2 datasets report 6.5× goodput improvement over vLLM and 4.7× over INFERCEPT.

**Compliance With Llm Reviewing Policy:**

Affirmed.

**Final Justification:**

(a) Fully resolved - My concerns have been adequately addressed. If you select this option, please consider adjusting your score accordingly.

**Key Questions For Authors:**

1. Why was LAMPS excluded from the experimental comparison? Is there a specific incompatibility?
2. How does performance degrade as prediction accuracy drops (e.g., to 40% bucket accuracy)?
3. What is the scheduling overhead when hundreds of requests are simultaneously in different stages?

**Limitations:**

The prediction accuracy limitation is acknowledged, but the missing LAMPS comparison and reliance on synthetic workloads are not adequately discussed.

**Strengths And Weaknesses:**

**Strengths:**

1. The space-time cost model — integral of memory occupancy over time across execution phases — is the right abstraction for this problem. It provides a principled way to compare heterogeneous request states and evaluate handling strategies (Preserve/Discard/Swap), and the value-density scheduling follows naturally.

2. The two-stage correction is well-motivated: prediction errors accumulate, and post-invocation correction using observed API call durations can significantly improve scheduling quality without requiring high Stage I accuracy. This sidesteps the prediction-accuracy problem effectively.

3. The evaluation covers a useful range of hardware (RTX 4090, H800, A100) and model sizes (6B to 70B). The ablation is informative — scheduling contributes more than the dynamic token budget (TTFT: 306s → 14.5s with scheduling alone vs. 293s with budget alone).

**Weaknesses:**

**Major:**

1. **Missing LAMPS baseline.** I was surprised not to see LAMPS (arXiv:2410.18248, Oct 2024) as a baseline — it's the most directly comparable work: it also predicts handling strategies, ranks by memory-over-time consumption, and achieves 27–85% latency improvement on vLLM. AugServe cites LAMPS in related work but does not compare against it experimentally. The 6.5× over vLLM and 4.7× over INFERCEPT numbers would be much more convincing alongside a LAMPS comparison. If LAMPS cannot be reproduced (e.g., code unavailability or vLLM version incompatibility), the authors should explain why and provide an alternative prediction-based scheduling baseline.

2. **Prediction accuracy and sensitivity.** The prediction module achieves only 65% bucket accuracy on the INFERCEPT dataset (Section 5). I didn't find any sensitivity analysis for prediction error. If Stage I ordering is substantially wrong, some requests may cause head of line blocking before Stage II kicks in. A controlled experiment injecting calibrated noise (e.g. reducing accuracy to 50% and 40%) would clarify whether degradation is graceful or exhibits a cliff.

3. **No formal guarantees for the scheduling policy.** The value-density greedy approach is motivated by knapsack intuition, but no approximation ratio or competitive ratio is provided. Recent work has established formal guarantees for LLM inference scheduling under memory constraints (e.g., arXiv:2504.11320). Positioning the heuristic within this landscape — even to argue that the problem structure precludes standard guarantees — would sharpen the contribution.

**Minor:**

4. **Synthetic workloads only.** Experiments use Poisson and Gamma arrival processes. Real tool-calling workloads exhibit bursty arrivals, correlated API patterns, and multi-round conversations. Production trace data would be more convincing.

5. **Scale limited to 4 GPUs.** Distributed settings with tensor/pipeline parallelism across many GPUs are not explored.

6. **Scheduling overhead under load.** The paper reports per-decision latency but does not characterize how overhead scales with the number of concurrent requests, particularly when many are in various paused states.

---

> ### Author Rebuttal · Authors · 2026-03-31
>
> Thank you for your thoughtful review.
> We appreciate your feedback and address your concerns below. All experiments are conducted on an H800 GPU with OPT-13B model on the INFERCEPT dataset.
> ### **W1 and Q1. Missing LAMPS baseline**
> Thank you for raising this important question.
> We clarify that LAMPS (arXiv 2024) was later published as MARS (NeurIPS 2025), which we already include in our evaluation (Appendix D.5, Fig.13–14), covering this line of work.
>
> As shown in Appendix D.5, AugServe consistently outperforms MARS with higher Goodput and lower latency, especially under high load. MARS adopts a per-round static cost model, treating each execution round independently, which becomes less effective under dynamic workloads. In contrast, AugServe models cross-round execution states and refines scheduling using runtime feedback after tool returns, which is critical in multi-stage execution.
> ### **W2 and Q2. Prediction accuracy and sensitivity**
> Thank you for raising this question.
>
> (1) Large Stage I errors could introduce temporary HoL blocking (e.g., a long request mispredicted and assigned a $Preserve$ policy). However, such cases are unlikely because: (a) they require compound mispredictions in both call duration and context growth, and (b) AugServe’s space-time cost–based policy selection naturally discourages $Preserve$ for long-running requests.
>
> (2) We evaluate prediction sensitivity in Appendix D.3 (Fig. 12) by injecting 10%–50% noise into both output length and external call duration predictions. The results show stable Goodput and latency with no cliff-like degradation.
> This is because predictions act as coarse priors, while feedback-driven scheduling corrects errors online and prevents accumulation.
> ### **W3. No formal guarantees for the scheduling policy**
> Thank you for this valuable suggestion.
> Ao et al. (arXiv:2504.11320) provide a strong theoretical framework for memory-constrained LLM inference.
> However, their analysis assumes continuous execution, monotonic memory growth, and mainly considers the $Discard$ policy with recomputation.
> These assumptions do not hold in augmented LLM inference, which is multi-stage and discontinuous due to tool calls, with non-monotonic memory usage and varing costs under different context-handling policies ($Preserve/Swap/Discard$).
> As a result, deriving formal guarantees for the full AugServe setting is significantly challenging.
>
> Under a simplified per-iteration setting with fixed value and memory cost, the problem reduces to a 0–1 knapsack, where our value-density greedy policy achieves a 1/2-approximation to the optimum.
> The full proof is provided in https://github.com/no-user-name123/ICML26-rebuttal and will be included in the appendix.
> ### **W4. Synthetic workloads only**
> We agree that real tool-calling workloads are bursty, correlated and multi-round.
>
> (1) We model burstiness using Gamma arrivals with varying coefficients of variation in Appendix D.4 (Table 4) and observe stable performance.
>
> (2) In addition, to capture variability in tool behaviors, we inject large perturbations (2×–3×) to return lengths and call durations:
> |Metric|Req/s|vLLM |InferCept|AugServe|
> |-|-|-|-|-|
> |Goodput (req/s)|3.0|0.10|0.07|0.98|
> ||4.0|0.09|0.14|0.84|
> |Avg-TTFT (s)|3.0|211.56|482.47|5.01|
> ||4.0|273.59|448.59|4.40|
>
> AugServe remains robust and consistently outperforms baselines.
> We agree that production traces would strengthen the results and will enrich our workload design to better approximate real-world scenarios.
> ### **W5. Scale limited to 4 GPUs**
> AugServe operates at the request scheduling layer and is largely orthogonal to tensor/pipeline parallelism, which affects per-request execution rather than scheduling decisions. Extending to larger distributed settings mainly requires adapting the resource model, while the core scheduling remains unchanged.
> We will include this as future work.
> ### **W6 and Q3. Scheduling overhead under load**
> We thank the reviewer for this important question.
>
> (1) We first report the average per-iteration scheduling overhead (s) under different request rates:
> |Req/s|InferCept|AugServe|
> |-|-|-|
> |3.0|0.013|0.018|
> |4.0|0.015|0.022|
>
> AugServe incurs slightly higher per-iteration overhead than InferCept due to richer scheduling logic, but remains small (18–22 ms per iteration).
>
> (2) We further analyze overhead (s) as the number of paused requests increases:
> |Paused Requests|AugServe|
> |-|-|
> |~50|0.002|
> |50–100|0.008|
> |100–150|0.016|
> |150–200|0.025|
> |≥200|0.021|
>
> Overhead slightly grows with the number of paused requests, but remains at the millisecond level even with hundreds of paused requests, indicating that scheduling does not become a bottleneck under high concurrency.
> This behavior is expected, as the scheduler is dominated by sorting and greedy selection, leading to smooth scaling with the number of active requests.
>
> Overall, AugServe maintains stable and efficient scheduling under high concurrency. We will include this analysis in the revision.

---

> > ### Author Rebuttal · Reviewer_pi6b · 2026-04-02
> >
> > I thank the authors for the detailed response. I now see that LAMPS and MARS are the same line of work, and MARS is already compared in Appendix D.5, so my concern about the missing baseline is resolved. The noise-injection experiments and the overhead measurements under high concurrency are reassuring. On formal guarantees, I appreciate the honest discussion of why the full setting resists standard analysis, and the 1/2-approximation for the simplified knapsack is useful; I look forward to seeing the proof in the appendix. It may also be worth clarifying how the problem structure here differs from related formulations under different infrastructure assumptions, so readers can see when existing guarantees apply and when they do not. The single-node scale limitation is minor given the overall evaluation scope. The rebuttal has addressed most of my concerns, and I am raising my score to 5. I hope the approximation proof and the discussion of related formulations make it into the camera-ready.

---

> > > ### Author Response · Authors · 2026-04-02
> > >
> > > Thank you for the constructive feedback and for taking the time to re-evaluate our work. We are glad that the clarifications on the LAMPS/MARS baseline, robustness analysis, and scheduling overhead addressed your concerns.
> > >
> > > We appreciate your suggestion regarding clarifying the differences between our problem setting and related formulations under different infrastructure assumptions. We will incorporate this discussion, along with the full approximation proof, in the camera-ready version.

---

### Official Review · Reviewer_ZA98 · 2026-03-10

**Soundness:** 3
**Presentation:** 3
**Significance:** 2
**Originality:** 2
**Overall Recommendation:** 4
**Confidence:** 3

**Summary:**

This paper investigates the scheduling challenges in LLM inference serving specifically within the context of tool-augmented workloads. The authors identify two primary limitations of current systems: (1) scheduling agnosticism regarding external call durations, which triggers severe Head-of-Line (HoL) blocking and SLO violations; and (2) fixed batching mechanisms that fail to optimize throughput under the dynamic execution patterns of external calls. To address these, the paper proposes AugServe, a framework that employs a dual-strategy approach. First, it implements a state-aware scheduler that incorporates predicted output lengths and external call latencies. Second, it introduces a dynamic batch-level token budget that adjusts based on real-time GPU memory availability and reclaimable memory from preemptible paused requests. Extensive evaluations demonstrate that AugServe significantly outperforms baselines in effective throughput and SLO attainment.

**Compliance With Llm Reviewing Policy:**

Affirmed.

**Final Justification:**

The authors' rebuttal addressed most of my concerns, and I believe this work meets the bar for publication.

**Key Questions For Authors:**

- In the motivation, the authors state that paused requests cause HoL blocking. However, isn't it possible that pausing a long-running request for an external call actually provides an opportunity to finish shorter requests earlier (effectively an involuntary SJF), thereby improving average job completion time? Could you provide a more detailed analysis of when this blocking occurs?
- In Section 4, the model treats $P_{i,k}^{ret}$ and $R_{i,k}^{res}$ as separate entities. However, under the "Discard" policy, context resumption and tool-return incorporation are typically handled in a single prefill pass. Why does the model separate them, and does this reflect the actual implementation?
- Could the authors clarify the scheduling frequency? Does the scheduler run at every iteration, or is it only triggered upon the arrival of new requests or the return of external calls?
- How were the SLOs in the experiments determined? Are they based on specific industry standards or adapted from previous literature in LLM serving?

**Limitations:**

The discussion on limitations is not included, but I guess it's not a fatal issue.

**Strengths And Weaknesses:**

Strengths:\
[+] The paper addresses a timely and significant problem: the scheduling complexity introduced by external tool calls in augmented LLM inference. \
[+] The motivation is well-grounded. The authors correctly identify that existing systems fail to exploit the unique memory reclamation opportunities presented by paused requests during external calls. \
[+] The proposed system, AugServe, demonstrates substantial performance gains over baselines like vLLM and INFERCEPT in terms of effective throughput.

Weakness:\
[-] The issue of a fixed batch-level token budget (Challenge 2) is not unique to augmented LLMs. General LLM serving systems already employ iteration-level scheduling and dynamic memory management to handle varying sequence lengths. While the authors incorporate "reclaimable memory" from paused requests, the conceptual leap from existing work (e.g., Orca, Past-future scheduler) appears incremental. The complexity added by external calls seems to be treated as a simple additive memory term rather than a fundamental architectural shift. \
[-] While the paper defines a mechanism to determine the total token capacity, it lacks a detailed explanation of how this budget is translated into the selection of specific requests within a batch. Specifically, the interplay between the scheduling priority and the hard constraints of the dynamic token budget needs further clarification to ensure no memory fragmentation or OOM occurs. \
[-] The system's efficacy relies heavily on the accuracy of predicting both output lengths and external call durations. \
    (i) For output length prediction, the fine-tuned BERT approach is traditional and its robustness is questionable given the inherent stochasticity of LLMs.\
    (ii) Predicting external call durations is even more challenging due to network latency and tool-side variability. Relying on pre-labeled datasets for these predictions limits the practical applicability of the system. The paper lacks a sensitivity analysis on how prediction errors impact overall system performance.

---

> ### Author Rebuttal · Authors · 2026-03-31
>
> Thank you for the thoughtful feedback. We appreciate the recognition of the problem and empirical improvements. Below, we clarify your questions on token budgeting and prediction, and will incorporate them into the revision.
> ### **W1. The design of token budgeting**
> Thank you for the insightful comment. Iteration-level scheduling and dynamic memory management have been explored in previous work.
> However, prior work addresses simple settings where memory usage evolves monotonically with sequence length.
> In augmented LLM inference, external calls introduce multi-stage execution, where memory usage and recomputation costs vary across stages, making previous work inapplicable.
>
> AugServe therefore models memory as a stateful, time-varying resource for augmented LLM inference, and integrates it into a unified space–time cost and scheduling formulation.
> Meanwhile, reclaimable memory in AugServe is not an additive extension of free memory, but depends on context-handling policies ($Preserve/Swap/Discard$) and future execution, requiring joint modeling in scheduling decisions, which is not captured in prior approaches.
> We will clarify this distinction in the revision.
> ### **W2. Interaction between scheduling and token budget**
> We thank the reviewer for the suggestion.
> In AugServe, the dynamic token budget is derived from available GPU memory and serves as a hard admission constraint.
>
> Requests are ranked by value density and admitted greedily, subject to the remaining budget.
> When a request requires tokens exceeds the budget,  only the fitting portion is admitted (similar to chunked prefill), and the rest is deferred to future iterations, ensuring bounded per-iteration work.
> Thus, admitted tokens always stay within memory limits, preventing OOM.
> Meanwhile, memory management relies on the underlying block-based allocation and is robust to fragmentation.
>
> In summary, scheduling priority determines which requests to serve, while the token budget strictly controls how much can be served, ensuring safe and efficient execution.
> We will clarify this interaction in the revision.
> ### **W3. Dependence on prediction**
> We thank the reviewer for this important concern.
> AugServe is robust to prediction errors due to its feedback-driven design, which continuously refines initial estimates using runtime information.
>
> (1) **Robustness**. Across diverse workloads, AugServe consistently achieves strong performance gains, demonstrating robustness to prediction imperfections.
> Predictions serve as coarse priors and are continuously refined via runtime feedback (e.g., realized return lengths and execution states), making the system less sensitive to predictor choice and preventing error accumulation.
>
> (2) **Sensitivity**. We provide a sensitivity analysis in Appendix D.3 (Figure 12), where 10%–50% noise is injected into predictions. The results show that Goodput and latency remain largely stable under such errors.
>
> We will highlight these points in the revision.
> ###  **Q1. HoL blocking under paused requests**
> We thank the reviewer for this insightful question.
> We agree that pausing a long-running request can allow shorter requests to finish earlier,  but only when the pause releases resources (e.g., under  $Discard$ or $Swap$).
>
> However, under the $Preserve$ policy, HoL blocking occurs because paused requests continue to occupy GPU memory. In particular, the KV cache remains resident, reducing available capacity and blocking subsequent requests despite the pause.
>
> Therefore, scheduling effectiveness should depends on the memory reclaimability of the context-handling policy, rather than request length alone.
> ### **Q2. Modeling of resumption and return tokens**
> We thank the reviewer for the clarification.
> The behaviors of context resumption and tool-return incorporation differ significantly across policies:
>
> (1) Under $Preserve$, resumption cost is negligible, and tool-return processing corresponds to incremental prefill.
>
> (2) Under $Swap$, resumption is dominated by KV cache transfer between CPU and GPU, which differs from return-token incorporation.
>
> (3) Only under $Discard$ are resumption and return-token incorporation fused into a single prefill pass.
>
> Therefore, this separation is a modeling abstraction to enable a unified formulation across policies and does not imply separate execution. We will clarify this in the revision.
> ### **Q3. Scheduling frequency**
> In AugServe, scheduling is performed at every iteration (iteration-level scheduling), where all active requests are re-evaluated to form the batch.
> ### **Q4. SLO determination**
> We thank the reviewer for the question.
> The SLOs follow commonly adopted settings in prior LLM serving literature (Section 6 line 375-377). TTFT < 1s reflects responsiveness, while normalized latency captures per-token efficiency, providing a balanced measure of responsiveness and throughput.
> We will clarify this more explicitly in the revision.

---

> > ### Author Rebuttal · Reviewer_ZA98 · 2026-04-01
> >
> > I thank the authors for their responses, and most of my previous concerns have been effectively addressed. However, I still have two remaining reservations. First, the explanation regarding the robustness of length prediction remains somewhat unconvincing. Given the inherent stochasticity of LLM generation (e.g., due to temperature settings), it is unclear how the proposed method maintains its accuracy across different sampling configurations. Second, regarding the scheduling overhead, I am concerned about the cumulative latency since the algorithm is invoked at every iteration. Despite these points, the overall improvements are notable, and I am willing to raise my score.

---

> > > ### Author Response · Authors · 2026-04-02
> > >
> > > We thank the reviewer for the positive feedback and for raising these important follow-up questions. We clarify both concerns below.
> > > ### **Q1.Robustness under stochastic generation**
> > > We agree that LLM output lengths are inherently stochastic (e.g., due to temperature), and precise prediction is fundamentally challenging. However, AugServe does not rely on accurate point estimates. Instead, it only requires coarse-grained length ranges to guide initial prioritization and continuously refines them using runtime feedback.
> > >
> > > First, output length is predicted using discretized buckets (50-token granularity), which makes the prediction robust to sampling variability. Variations induced by different decoding configurations may shift predictions across buckets, but such shifts typically lead to only coarse changes in estimated cost, resulting in limited impact on the relative ordering of requests.
> > >
> > > In other words, the scheduling decision is largely invariant to small stochastic fluctuations. In our experiments (Section 6.1 Fig. 7), AugServe consistently achieves strong performance across diverse settings (hardware, models, and workloads), indicating that such coarse-grained estimates are sufficient to guide effective scheduling.
> > >
> > > Second, and more importantly, AugServe is inherently feedback-driven. As described in Section 5.3, predictions are only used in the pre-invocation stage. Once the external call returns, scheduling decisions are fully corrected using realized return lengths and execution states. This prevents prediction errors from accumulating across segments.
> > >
> > > We further validate this via noise injection experiments (Appendix D.3, Fig. 12), where both output lengths and call durations are perturbed by up to 50%. The results show that Goodput and normalized latency remain largely stable, demonstrating that AugServe is insensitive to substantial prediction noise. We will clarify that this robustness is able to extend to stochastic decoding settings.
> > >
> > > ### **Q2. Scheduling overhead under iteration-level invocation**
> > > We thank the reviewer for this important question. We understand the concern that invoking the scheduler at every iteration may introduce cumulative overhead. However, the overhead introduced by AugServe is small and does not become a bottleneck, while the resulting latency and throughput gains are substantial, as we detail below.
> > >
> > > **(1) Scheduling overhead analysis.**
> > >
> > > First, we report the average per-iteration scheduling overhead (s) under different request rates:
> > >
> > > |Req/s|InferCept|AugServe|
> > > |-|-|-|
> > > |3.0|0.013|0.018|
> > > |4.0|0.015|0.022|
> > >
> > > AugServe incurs slightly higher per-iteration overhead than InferCept due to richer scheduling logic, but remains small (18–22 ms per iteration).
> > >
> > > Second, we further examine how scheduling overhead scales with the number of paused requests:
> > >
> > > |Paused Requests|Scheduling Overhead (s)|
> > > |-|-|
> > > |~50|0.002|
> > > |50–100|0.008|
> > > |100–150|0.016|
> > > |150–200|0.025|
> > > |≥200|0.021|
> > >
> > > The overhead increases smoothly with the number of paused requests and remains at the millisecond level even with hundreds of concurrent requests. This indicates that the scheduler scales well under high concurrency and does not exhibit any abrupt growth or bottleneck behavior.
> > >
> > > **(2) Scheduling performance gains analysis**.
> > >
> > > Iterative scheduling enables the system to promptly re-evaluate and re-prioritize requests in the next iteration after state changes (e.g., tool returns), allowing newly runnable or more efficient requests to be scheduled earlier than waiting behind stalled ones. This reduces queueing delay and mitigates HoL blocking.
> > >
> > > In absolute terms, the scheduling overhead is on the order of milliseconds per iteration (≈18–22 ms), whereas the reduction in queueing delay can reach hundreds of seconds in TTFT under high load (e.g., reduced from hundreds of seconds to tens of seconds in Section 6 Fig. 7 and Table 2).
> > >
> > > This orders-of-magnitude difference shows that the benefit of improved scheduling decisions overwhelmingly outweighs the modest per-iteration overhead.
> > >
> > > Therefore, even with scheduling invoked in every iteration, the scheduling overhead remains small while significantly improving system performance.
> > >
> > > We will include these measurements and clarify this point in the revision.

---

### Official Review · Reviewer_yinw · 2026-03-12

**Soundness:** 3
**Presentation:** 3
**Significance:** 3
**Originality:** 3
**Overall Recommendation:** 4
**Confidence:** 4

**Summary:**

Augmented LLMs frequently pause when invoking external tools, leading to severe Head-of-Line blocking and a sharp drop in system throughput.
This paper proposes the AugServe inference framework, which utilizes "state-aware adaptive scheduling" and "dynamic batch-level token budget" mechanisms to flexibly schedule requests and allocate memory based on their real-time execution states.
Experimental results demonstrate that this approach significantly reduces queuing delays and increases the system's effective throughput (goodput) to 6.5x and 4.7x that of existing systems like vLLM and INFERCEPT, respectively.

**Compliance With Llm Reviewing Policy:**

Affirmed.

**Key Questions For Authors:**

see weakness.

**Limitations:**

see weakness.

**Strengths And Weaknesses:**

### Strengths
1. **Significant Breakthroughs in Performance**\
a. **Multi-fold Increase in Throughput**: Achieved a 6.5x and 4.7x increase in effective throughput (goodput) compared to vLLM and INFERCEPT, respectively.\
b. **Drastic Reduction in Latency**: Reduced the average TTFT by 95.6% (compared to vLLM) and 96.0% (compared to INFERCEPT).
2. **Innovative Advantages in Architecture Design**\
a. **State-Aware Adaptive Scheduling**: The scheduler continuously and dynamically updates request priorities based on their execution states, space-time cost, and actual runtime feedback.\
b. **Dynamic Token Budget**: The system adapts the processing capacity in real-time based on currently available free GPU memory and the "reclaimable memory" released by paused requests, thereby maximizing hardware resource utilization.\
c. **Low-Overhead Prediction Module**: The auxiliary prediction model adds less than 0.1% overhead to the overall end-to-end system latency.
3. **Exceptional System Robustness and Generality**\
As verified in the appendix, the method demonstrates high tolerance to prediction errors, strong resilience against bursty traffic, and broad adaptability across various scenarios.

### Weaknesses
1. **Questionable Generalizability to Unseen Tools**\
The prediction model relies heavily on historical features.
Although the paper tests prediction errors ranging from 10% to 50%, it does not cover dynamically plugged-in new tools (zero-shot scenarios) or sudden delays caused by network fluctuations, leaving its generalizability in real-world environments unverified.
2. **OOM Risks Triggered by Long Returns**\
External calls in practical applications (e.g., RAG) can instantaneously return thousands of tokens, whereas Figure 4 in the paper only tests extremely short mixed return lengths of 32 and 256 tokens. It remains questionable whether the existing dynamic token budget can withstand the OOM shock caused by the sudden injection of a massive number of tokens.
3. **Lack of Evaluation for Complex Serial Calls** \
The experiments primarily test macro-level workloads under different arrival distributions. Although the system models multiple service stages, it lacks a detailed latency breakdown for complex agents (e.g., the ReAct framework) during multiple serial "think-and-call" loops within a single request.

---

> ### Author Rebuttal · Authors · 2026-03-31
>
> We thank the reviewer for recognizing the innovative system design and strong performance of our work. We also appreciate the thoughtful questions on generalizability and robustness. We address these points below. All experiments are conducted on an H800 GPU with OPT-13B model on the INFERCEPT dataset.
> ### **Q1. Questionable Generalizability to Unseen Tools**
> We thank the reviewer for raising this important concern.
> We evaluate robustness to unseen tools and unpredictable behaviors by simulating such scenarios via substantial perturbations to both tool return lengths and call durations. Specifically, we scale return lengths by a random factor in [0.5, 2] and call durations in [0.3, 3], modeling zero-shot tools and network fluctuations.
> The results are shown below:
> |Metric|Req/s|vLLM |InferCept|AugServe|
> |-|-|-|-|-|
> |Goodput (req/s)|3.0|0.10|0.07|0.98|
> ||4.0|0.09|0.14|0.84|
> |Avg-TTFT (s)|3.0|211.56|482.47|5.01|
> ||4.0|273.59|448.59|4.40|
>
> AugServe remains highly robust under these perturbations. It consistently outperforms baseline systems (vLLM and InferCept) in both Goodput and TTFT, maintaining stable performance even under challenging conditions.
> This robustness stems from AugServe’s adaptive, state-aware scheduling design (Section 5.3).
> Predictions guide scheduling at the granularity of each service segment, rather than serving as fixed estimates for the entire request.
> Specifically, (1) an initial scheduling priority is estimated upon request arrival, and (2) after each tool return, priorities are continuously refined using runtime feedback. This pre-invocation estimation and post-invocation refinement allow the scheduler to adapt online even under unseen tool behaviors.
> We will include this analysis in the revision.
> ### **Q2. OOM Risks Triggered by Long Returns**
> We appreciate this concern and further analyze OOM risks under long returns.
> AugServe can prevent sudden memory spikes and ensure that memory usage remains within device capacity, even under extremely long return sequences. This is ensured by its design: (1) Scheduling is performed under a strict dynamic token budget computed from available and reclaimable GPU memory, enforcing a hard bound on per-iteration memory usage; and (2) returned tokens are incorporated incrementally, where only the budget-allowed portion is processed per iteration, and the remainder is deferred to subsequent iterations.
>
> To illustrate this, we further evaluate robustness under two load levels (3.0 and 4.0 req/s), comparing normal (2k–3k) and extreme (8k–10k) return lengths:
> |Metric|Req/s|Return|vLLM|InferCept|AugServe|
> |-|-|-|-|-|-|
> |Goodput (req/s)|3.0|2k–3k|0.27|0.33|1.26|
> |||8k–10k|0.20|0.30|1.26|
> ||4.0|2k–3k|0.17|0.28|1.31|
> |||8k–10k|0.15|0.24|1.17|
> |Avg-TTFT (s)|3.0|2k–3k|77.33|137.56|13.17|
> |||8k–10k|121.49|194.81|10.92|
> ||4.0|2k–3k|186.90|309.60|2.71|
> |||8k–10k|217.88|338.31|5.82|
>
> Across all stress tests, we observe no OOM or memory-related failures. While baseline systems (vLLM and InferCept) degrade under long returns (lower Goodput and higher TTFT),
> AugServe consistently outperforms them, with only minor fluctuations under extreme conditions. We will include these results in the revision.
> ### **Q3. Lack of Evaluation for Complex Serial Calls**
> We thank the reviewer for this insightful suggestion.
> We further analyze latency breakdown at the single-request level for complex multi-hop executions.
>
> Specifically, we perform think–action segmented latency analysis over ~2,500 requests, grouping them by the number of internal think–action loops (1, 2–3, ≥4), and decomposing end-to-end latency into queueing, model execution, and tool-call time:
>
> |Calls|System|Queuing (s)|Execute (s)|Tool (s)|Total (s)|
> |-|-|-|-|-|-|
> |1|vLLM|340.6|34.76|7.20|382.65|
> ||InferCept|401.93|24.99|6.44|433.35|
> ||AugServe|37.42|21.73|25.62|84.78|
> |2–3|vLLM|432.47|60.25|25.46|518.17|
> ||InferCept|396.71|44.23|26.31|467.25|
> ||AugServe|41.03|23.95|40.53|105.52|
> |≥4|vLLM|575.78|53.61|42.70|672.09|
> ||InferCept|391.40|37.38|73.29|502.08|
> ||AugServe|61.19|28.31|86.14|175.64|
>
> As the number of serial calls increases, the baseline systems (vLLM and InferCept) exhibit rapidly growing queueing delays, reflecting severe HoL blocking. In addition, their execution time is also higher than AugServe, indicating inefficiencies of context resumption.
> In contrast, AugServe consistently maintains lower queueing and execution time across all call-count buckets, demonstrating robust scheduling under multi-stage execution.
> Tool-call time reflects external latency and is not directly optimized by scheduling; its variation mainly arises from differences in the composition of completed requests.
> Therefore, the reduction in end-to-end latency primarily comes from mitigating queueing and resumption inefficiencies.
>
> In summary, AugServe improves not only macro-level throughput, but also efficiency within individual complex requests with multiple serial think–action loops. We will include this analysis in the revision.

---

### Official Review · Reviewer_bGau · 2026-03-13

**Soundness:** 3
**Presentation:** 3
**Significance:** 3
**Originality:** 3
**Overall Recommendation:** 4
**Confidence:** 3

**Summary:**

The paper introduces AugServe, an inference serving framework specifically designed for Augmented Large Language Models (LLMs). These models improve performance by invoking external calls (e.g., APIs or databases), which introduces "execution heterogeneity" because requests must pause while awaiting external responses. Traditional systems like vLLM and INFERCEPT suffer from Head-of-Line (HoL) blocking and inefficient resource use under these workloads.

AugServe addresses these challenges through two primary innovations:

1. State-Aware Request Scheduling: A greedy value-density scheduling policy that explicitly models the multi-stage lifecycle of augmented requests. It uses a space-time cost model to prioritize requests based on their current execution state (prefill, decode, tool-wait, or resumption) and actual runtime feedback.

2. Dynamic Batch-Level Token Budget: A mechanism that adapts the number of tokens processed per iteration based on both free GPU memory and reclaimable memory from paused requests.


Evaluation across various models (e.g., GPT-J-6B, Llama-3-70B) and datasets (ToolBench, INFERCEPT) shows that AugServe achieves up to 6.5x higher effective throughput (goodput) than vLLM and 4.7x higher than INFERCEPT, while significantly reducing latency.

**Compliance With Llm Reviewing Policy:**

Affirmed.

**Final Justification:**

The rebuttal addressed my main concerns, I lean toward accepting this submission.

**Key Questions For Authors:**

1. How sensitive is the system to hyperparameter $\beta$ (Eq. 2)?

2. How does the design in this paper interact with other inference optimizations such as speculative decoding and chunked prefill?

**Limitations:**

yes

**Strengths And Weaknesses:**

### **Strengths**

- State-aware cost modeling is reasonable and intuitive. Unlike standard Shortest-Job-First (SJF) models that rely only on request length, AugServe’s space-time cost model accounts for the memory residency of paused requests and the varied costs of context resumption (e.g., recomputation vs. swapping).

- The introduction of "reclaimable memory" into the token budget calculation is novel and it is a significant improvement over prior systems that only consider free GPU memory.

- The empirical results are strong. The system demonstrates dramatic improvements in goodput and Time-to-First-Token (TTFT) across multiple hardware configurations and complex tool-use datasets.


* * *

### **Weaknesses**

- Lack of sensitivity discussion on hyperparameter $\beta$ . The scheduling value ($V_i^e$) depends on a hyperparameter ($\beta$) that balances throughput and delay awareness. The paper could benefit from more discussion on how sensitive the results are to different choices of $\beta$  across varied workloads.

---

> ### Author Rebuttal · Authors · 2026-03-31
>
> Thank you for recognizing the novelty of our state-aware scheduling design and the strong empirical results. We appreciate your insightful questions regarding $\beta$ sensitivity and the interaction with other inference optimizations. Below, we provide additional clarifications.
>
> ### **Q1. How sensitive is the system to hyperparameter $\beta$ (Eq. 2)?**
> We thank the reviewer for the thoughtful question. We provide additional experiments and analysis to show that the system is not overly sensitive to $\beta$.
>
> We conduct sensitivity analysis on an H800 GPU with OPT-13B model on the INFERCEPT dataset. In the paper, we use $\beta = 5e-5$. Here, we evaluate a wider range $\beta \in \lbrace 1e-3, 1e-4, 5e-5, 1e-5, 1e-6 \rbrace$ under different workloads.
> The results are summarized below:
>
> |Metric|Req/s|$\beta$=1e-3|$\beta$=1e-4|$\beta$=5e-5 * |$\beta$=1e-5| $\beta$=1e-6|
> |-|-|-|-|-|-|-|
> |Goodput (req/s)|3.0|1.01|1.12|1.07|1.04|1.02|
> ||5.0|0.82|0.94|0.90|0.94|0.86|
> |P99-TTFT (s)|3.0|276.02|109.05|98.66|111.89|278.10|
> ||5.0|298.65|152.84|157.45|166.99|226.58|
>
> (1) **Robustness.** AugServe is not overly sensitive to $\beta$. Across workloads, the best performance consistently falls within a stable range around $\beta \approx$ 1e-5-1e-4, rather than a single sharply tuned value.
> The default value used in the paper is $\beta$  = 5e-5, falling within this range, and achieves stable performance across workloads (Section 6).
>
> (2) **Large $\beta$.** When $\beta$ is too large (e.g., 1e-3), the scheduler over-emphasizes waiting time, prioritizing long-waiting or paused requests at the cost of execution efficiency, leading to worse tail latency and lower Goodput.
>
> (3) **Small $\beta$.** When $\beta$ is too small (e.g., 1e-6), the scheduler behaves closer to a throughput-dominant greedy policy, which degrades tail latency and can reduce Goodput under higher load.
>
> In summary, $\beta$ remains effective over a relatively wide range, maintaining strong performance without requiring fine-grained tuning for specific workloads.
> We will include the complete $\beta$ sensitivity analysis in the final version.
>
> ### **Q2. How does the design in this paper interact with other inference optimizations such as speculative decoding and chunked prefill?**
> We thank the reviewer for raising this important question. AugServe is a scheduling layer that operates on top of the execution engine and is orthogonal to existing inference optimizations such as speculative decoding and chunked prefill.
>
> (1) Speculative decoding is complementary to AugServe. Speculative decoding accelerates token generation within each request, while AugServe determines which requests to schedule. Therefore, the two reinforce each other: speculative decoding improves the efficiency of each request, while AugServe achieves higher system-level throughput and lower latency through better scheduling.
>
> (2) Chunked prefill is naturally supported by AugServe. Chunked prefill handles long prompts by splitting large prefills into smaller segments. AugServe naturally supports this behavior through token-budget-based scheduling: when a request’s prefill exceeds the per-iteration token budget, only the budget-allowed portion is processed, and the remaining tokens are deferred to subsequent iterations. This leads to incremental prefill execution that aligns well with chunked prefill.
>
> Overall, these methods are complementary: inference optimizations enhance per-request execution, while AugServe improves system-level efficiency through better request scheduling.
> As a result, AugServe can be readily integrated into existing systems and combined with their optimizations.
> We will further elaborate on these interactions in the revision.

---

> > ### Author Rebuttal · Reviewer_bGau · 2026-04-06
> >
> > The authors have fully addressed my earlier concerns.
> >
> > The additional experiments—particularly the $\beta$‑sensitivity analysis—directly respond to the key questions raised in my initial review. The new results demonstrate that the system is not overly sensitive to the choice of $\beta$. The clarifications on the interactions between AugServe and other inference optimizations are reasonable.

---

> > > ### Author Response · Authors · 2026-04-06
> > >
> > > We sincerely thank the reviewer for the thoughtful follow-up and for acknowledging that our responses have fully addressed the concerns. We are glad that the additional experiments, especially the $\beta$ sensitivity analysis, as well as the clarifications on the interactions with other inference optimizations, were helpful.
> > >
> > > We truly appreciate the reviewer’s time, careful evaluation, and constructive feedback throughout the process. We would be grateful for the reviewer’s consideration in the final evaluation.

---

### Decision · Program_Chairs · 2026-04-30

**Decision:**

Accept (regular)

**Comment:**

The paper addresses an important and increasingly relevant systems problem in augmented LLM serving, namely how to schedule requests efficiently when external tool calls introduce pauses, heterogeneous execution states, and dynamic memory pressure. Its main contribution is a state-aware scheduling framework coupled with a dynamic token-budget mechanism that accounts for reclaimable memory from paused requests; this is a technically coherent design and is supported by strong empirical gains in goodput and latency across multiple models, datasets, and hardware settings. The main concerns raised in review are about the dependence on prediction accuracy, the absence of the most relevant baseline, robustness to long tool returns and complex serial tool-use patterns, and the overhead of iteration-level scheduling. These concerns were substantially addressed in the rebuttal and follow-up discussion: the authors clarified the comparison to MARS, provided additional sensitivity and stress-test evidence, and showed that scheduling overhead remains modest relative to the large queueing reductions achieved. The remaining limitations, including reliance on synthetic workloads, lack of production traces, and the absence of stronger formal guarantees for the full scheduling problem, are real but not decisive. Taken together, the work is technically sound, well motivated, and useful to the ML systems community, and I therefore recommend acceptance.